# The algorithm of microphysical parameter profiles of aerosol and small cloud droplets based on the dual wavelength Lidar data

Huige Di, Xinhong Wang, Ning Chen, Jing Guo, Wenhui Xin, Shichun Li, Yan Guo, Qing Yan, Yufeng Wang, Dengxin Hua*

School of Mechanical and Precision Instrument Engineering, Xi'an University of Technology, Xi'an 710048, China

*Correspondence to*: Dengxin Hua (dengxinhua@xaut.edu.cn)

**Abstract.** This study proposed an inversion method of atmosphere aerosol or cloud microphysical parameters based on dual wavelength lidar data. The matching characteristics between aerosol/cloud particle size distribution and Gamma distribution were studied using aircraft observation data. The feasibility of particle effective radius retrieval from lidar ratio and backscatter ratio was simulated and studied. A method for inverting the effective radius and number concentration of atmospheric aerosols or small cloud droplets using backscatter ratio was proposed, and the error sources and applicability of the algorithm were analyzed. This algorithm was suitable for the inversion of uniformly mixed and single property aerosol layers or small cloud droplets. Compared with the previous study, this algorithm could quickly obtain the microphysical parameters of atmosphere particles and has good robustness. For aerosol particles, the inversion range that this algorithm can achieve was 0.3-1.7 μm. For cloud droplets, it was 1.0-10 μm. An atmosphere observation experiment was conducted using the multi-wavelength lidar developed by Xi'an University of Technology, and a thin cloud layer was captured. The microphysical parameters of aerosol and cloud during this process were retrieved. The results clearly demonstrate the growth of effective radius and number concentration.

**Key words**: Lidar; Effective radius; Gamma distribution; Aerosol; Cloud

## 1 Introduction

The vertical characteristics of aerosol and cloud are of great significance for the study of many scientific issues, such as the interaction between aerosol and cloud, the mechanism of atmospheric pollution generation, and so on (Lohmann and Feichter, 2005; Kulmala et al., 2004; Miffre et al., 2010). The high-precision detection of aerosol and cloud microphysical parameters at vertical altitude is important. At present, the main methods for obtaining atmosphere aerosol or cloud microphysical parameters include in-situ observation (He et al., 2019; Moore et al., 2021; Gao et al., 2022a; Gao et al., 2022b) and remote sensing observation (Vivekanandan et al., 2020; Johnson et al., 2009). People can obtain microphysical parameters of cloud or aerosol at vertical altitudes by mounting in-situ observation instruments on equipment such as airplanes or balloons (Kaufman et al., 1998; Cai et al., 2022), but this method has a low detection frequency and cannot obtain continuous observation data with high temporal and spatial resolution (Zhao et al., 2018). Lidar, with its advantages of high temporal and spatial resolution and high detection sensitivity, has been widely used in the field of atmosphere detection, and has important application potential in detecting optical and microphysical parameters of atmosphere aerosol and cloud (Vivekanandan et al., 2020; Hara et al., 2018; Siomos et al., 2017; Kanitz et al., 2013; Dionisi et al., 2018).

The remote sensing detection of aerosol microphysical parameters mainly uses three wavelength lidar, which can obtain four or more optical parameters (usually requiring two extinction coefficients @355 nm&532 nm and three backscatter coefficients @ 355 nm&532 nm&1064 nm) for the retrieval of aerosol microphysical parameters (Veselovskii et al., 2004; Müller et al., 1999; Veselovskii et al., 2009). The regularization algorithm (Kolgotin et al., 2023; Veselovskii et al., 2002), the principal component analysis (PCA) technique (Martin et al., 2013), and the linear estimation algorithm (Veselovskii et al., 2012) have been used for determining the aerosol bulk properties. These algorithms do not require the assumption of complex refractive index or aerosol particle size distribution (APSD), so they have been widely studied, but their applications are limited. The inversion results are unstable, and there will be good results under certain spectral types; however, in some cases, the

inversion error is very large. Not only that, the above methods require the complex lidar hardware systems (Di et al., 2018a; Meskhidze et al., 2021; Müller et al., 2014). Therefore, the above algorithms cannot be well applied in most lidar systems (most lidars in AERONET are dual wavelength), and it is necessary to establish a more reasonable method for inverting microphysical parameters. For clouds, there are two methods used for the detection of cloud microphysical parameters. The first method is using lidar/radar synergy for cloud microphysical parameters (Wang et al., 2002; Vivekandan et al., 2020; Zhang et al., 2021), which can achieve the retrieval of cloud droplet with large cloud particles. For thin and sparse clouds or nascent clouds, cloud droplet particles are usually small and cannot be detected by millimeter wave cloud radar, which affects the application of this method. The second method is to use multiple scattering information in clouds detected by multi field of view (FOV) or dual FOV lidar to retrieve microphysical parameters of water clouds (Wang et al. 2022). However, in order to obtain multiple scattering signals using ground-based lidar, the larger FOV of telescope is required, which will greatly affect daytime detection.

This study proposes an inversion method of atmosphere aerosol or cloud microphysical parameters based on dual wavelength Lidar. This article mainly includes the following parts: in Section 2, we studied the APSD and cloud droplet size distribution (CDSD) measured by airborne instruments and found that they are basically consistent with the Gamma distribution, and extract the statistical characteristics of their Gamma distribution parameters; In Section 3, the inversion method and simulation analysis results were presented and described; In Section 4, an atmosphere observation result by lidar was presented; Section 5 is the conclusion and discussion.

## 2 Gamma distribution statistical characteristics of APSD and CDSD

### 2.1 Gamma distribution

The particle size distribution (Di et al., 2018a) is the variation of particle number with particle radius within a certain radius range $r \sim r + \mathrm{d}r$ per unit volume, defined as

$$n(r) = \frac{\mathrm{d}N}{\mathrm{d}r} \tag{1}$$

here, $r$ is the particle radius, $n(r)$ is the particle size distribution, $N$ is the total number of particles per unit volume. The effective radius (Di et al., 2018a) is an important parameter that characterizes the average particle size, defined as the ratio of the third-order and second-order moments of the particle size distribution, as shown below

$$r_{\mathrm{eff}} = \frac{\int_{r_{\min}}^{r_{\max}} r^3 n(r) \mathrm{d}r}{\int_{r_{\min}}^{r_{\max}} r^2 n(r) \mathrm{d}r} \tag{2}$$

The most common models for APSD are Junge distribution and lognormal distribution. CDSD usually described as Gamma distribution or corrected Gamma distribution (Kolgotin et al., 2023). The Gamma function has the advantages of integrability and recursion of various order functions. In this paper, the Gamma distribution is used to describe APSD and CDSD, and shown as

$$n(r)\mathrm{d}r = a r^b e^{-cr} \mathrm{d}r \tag{3}$$

here, $a$ is related to particle concentration, $b$ is a dimensionless parameter representing shape factor, which is related to spectral width, and $c$ is a slope parameter.

In mathematics, $\Gamma(x)$ is defined as Gamma function, and is as follow

$$\Gamma(x) = \int_0^{+\infty} t^{x-1} e^{-t} \mathrm{d}t \ , \ (x > 0) \tag{4}$$

Its p-th moment of Gamma distribution can be expressed as

$$M_p = \int_{r_{\min}}^{r_{\max}} r^p n(r) \mathrm{d}r = \int_{r_{\min}}^{r_{\max}} r^p a r^b e^{-cr} \mathrm{d}r = \frac{a}{c^{p+b+1}} \Gamma(p+b+1) \tag{5}$$

78      The effective radius requires second-order and third-order moments, which are

$$M_2 = \int_{r_{\min}}^{r_{\max}} r^2 n(r) \mathrm{d}r = \frac{a}{c^{2+b+1}} \Gamma(2+b+1) \tag{6}$$

$$M_3 = \int_{r_{\min}}^{r_{\max}} r^3 n(r) dr = \frac{a}{c^{3+b+1}} \Gamma(3+b+1) \tag{7}$$

81      Substituting Eq. (6) and Eq. (7) into Eq. (2) yields the effective radius as follow

$$r_{\text{eff}} = \frac{M_3}{M_2} = \frac{\int_{r_{\min}}^{r_{\max}} r^3 n(r) \mathrm{d}r}{\int_{r_{\min}}^{r_{\max}} r^2 n(r) \mathrm{d}r} = \frac{1}{c} \frac{\Gamma(3+b+1)}{\Gamma(2+b+1)} \tag{8}$$

83      The Gamma function has recursion, as shown in the following formula

$$\Gamma(x+1) = x\Gamma(x) \tag{9}$$

85      According to Eq. (8) and Eq. (9), the effective radius can be simplified as

$$r_{\text{eff}} = \frac{b+3}{c} \tag{10}$$

## 2.2 APSDs and CDSDs in the vertical altitude

     In order to study the characteristics of APSDs and CDSDs in the vertical altitude, the APSDs and CDSDs obtained from aircraft observations by the Hebei Provincial Weather Modification Office were analyzed (from 2005 to 2006). APSDs were measured by the PCASP-100X probe, and CDSDs were obtained by the FSSP-100-ER probe (Di et al., 2018b). The PCASP-100X is an optical particle counter for measuring aerosol size distribution from 0.10 μm to 3.00 μm in diameter in 15 different size bins with a frequency of 1 Hz. The sample flow volume in the PCASP-100X was set to 1 cm$^3$ s$^{-1}$. FSSP-100-ER is an instrument that measure cloud droplet size and concentration using light scattering, with the measurement range of 0.5-47 μm.

     The obtained APSDs and CDSDs were fitted one by one using Gamma function, and statistically analyze these fitting parameters. In order to minimize the error at all radius, the minimization problem is solved using the following equation

$$\int_{0}^{D_{\max}} \left( \log\left(f_m(D)\right) - \log\left(f_{fitted}(D)\right) \right)^2 \mathrm{d}D \rightarrow \min \tag{11}$$

here, $f_m(D)$ is the actual particle size distribution measured by the PCASP-100X, $f_{fitted}(D)$ is the fitted distribution, $D$ is the aerosol particle diameter, $D_{\max}$ is the measured maximum particle diameter. The goodness of fit $R^2$ is used to represent the difference between the fitting function and the measured data. The definition of goodness of fit is as follows::

$$R^2 = 1 - \frac{\sum_{i=1}^{n} \left(y_i - \hat{y}_i\right)^2}{\sum_{i=1}^{n} \left(y_i - \bar{y}\right)^2} \tag{12}$$

where $y_i$ is the measured value, $\hat{y}_i$ is the predictive value, $\bar{y}$ is the mean measured value. The numerator represents the sum of squared residuals, and the denominator represents the sum of squared total deviations.

     ~3500 sets of APSDs and 2221 sets of CDSDs were statistically analyzed. Over 95% of the data have a high goodness of fit in the Gamma distribution. The goodness of fit of CDSDs is higher than that of APSDs, with CDSDs of 0.983 and APSDs of 0.856. The parameter a of CDSDs are significantly larger than that of APSDs, and there are obvious differences of b and c for cloud and aerosol. The literature suggests that there is a certain functional relationship between the Gamma parameters b and c of CDSDs (Ding et al., 2023). Statistical analysis was conducted on the $b$ and $c$ parameters of APSDs and CDSDs, as shown in Fig. 1.

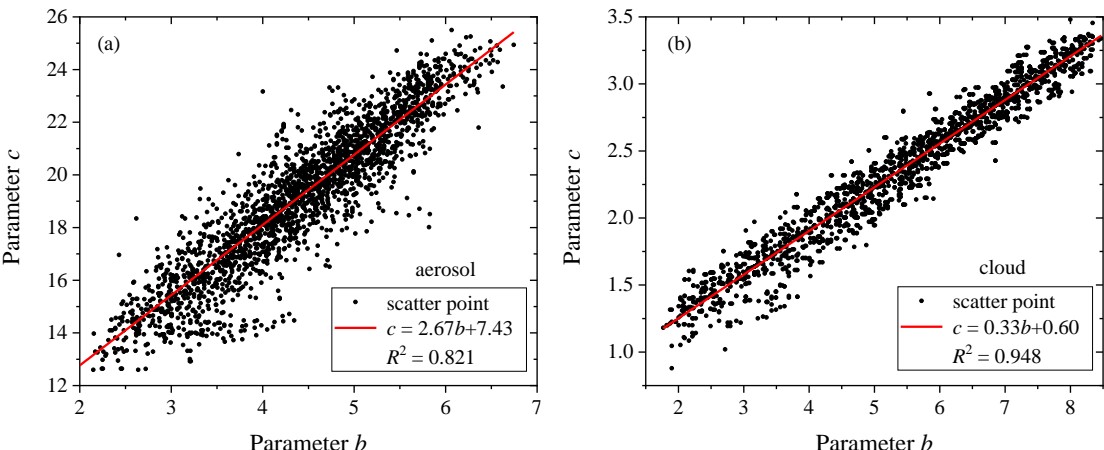

**Figure 1.** Statistical Results of parameter $b$ and $c$ in aerial survey data. (a)Aerosol particles, (b)cloud droplets.

According to Fig. 1, there are the remarkable linear relationships between parameter $b$ and $c$. The fitting functions for CDSDs and APSDs are as follows

$$\begin{cases} c_{\text{cloud}} = 0.33b_{\text{cloud}} + 0.60 \\ c_{\text{aerosol}} = 2.67b_{\text{aerosol}} + 7.43 \end{cases} \tag{13}$$

The linear relationship between the two parameters of CDSDs is better with a goodness of fit of 0.948, and a linear goodness of fit of 0.821 for APSDs. According to the statistical results, the parameter $b$ of APSDs at vertical height is mainly distributed in the range of 2-7, and CDSDs is mainly distributed in the range of 2-8.

## 3. The Inversion method for microphysical parameters of atmosphere aerosols or small cloud droplets

### 3.1 Inversion algorithm

The first step in this algorithm is the retrieval of the effective radius. The parameter $a$ in Gamma distribution shown in Eq. (4) is related to number concentration. The ratio $O_R(m, r)$ (lidar ratio or color ratio) of the two optical parameters can eliminate parameter $a$, and can be written as

$$O_R(m, r) = \frac{g_1(\lambda_1)}{g_2(\lambda_2)} \tag{14}$$

here, $m$ is the complex refractive index of particles, $g_1(\lambda_1)$ and $g_2(\lambda_2)$ are the optical parameters at two wavelengths $\lambda_1$ and $\lambda_2$, respectively. It can also be written as follows

$$O_R(m, r) = \frac{\int_{r_{\min}}^{r_{\max}} \pi r^2 Q_1(m, r, \lambda_1) r^b e^{-cr} \mathrm{d}r}{\int_{r_{\min}}^{r_{\max}} \pi r^2 Q_2(m, r, \lambda_2) r^b e^{-cr} \mathrm{d}r} \tag{15}$$

where $Q_1$ and $Q_2$ are the extinction efficiency factor or backscattering efficiency factor at $\lambda_1$ and $\lambda_2$. Using the effective radius in the Eq. (10) instead of parameter $c$, the Eq. (13) can be written as follows

$$O_R(m, r_{\text{eff}}) = \frac{\int_{r_{\min}}^{r_{\max}} \pi r^2 Q_1(m, r, \lambda_1) r^b e^{-\frac{b+3}{r_{\text{eff}}}r} \mathrm{d}r}{\int_{r_{\min}}^{r_{\max}} \pi r^2 Q_2(m, r, \lambda_2) r^b e^{-\frac{b+3}{r_{\text{eff}}}r} \mathrm{d}r} \tag{16}$$

According to the Eq. (11) and Eq. (14), if the ratio of optical parameters monotonically changes with the effective radius, the effective radius can be obtained from the ratio of optical parameters, and then parameters $b$ and $c$ can also be obtained according to Eq. (11). The ratio here can be chosen as the ratio of backscatter or extinction coefficient of two wavelengths (color ratio) or the ratio of extinction coefficient of one wavelength to backscatter coefficient (lidar ratio).

After obtaining $b$ and $c$, $a$ can be derived from the Eq. (15), written as

$$a = \frac{\int_{r_{\min}}^{r_{\max}} \pi r^2 Q_1(m,r,\lambda_1) a r^b e^{-cr} \mathrm{d}r}{\int_{r_{\min}}^{r_{\max}} \pi r^2 Q_1(m,r,\lambda_1) r^b e^{-cr} \mathrm{d}r} = \frac{g_1(\lambda_1)}{\int_{r_{\min}}^{r_{\max}} \pi r^2 Q_1(m,r,\lambda_1) r^b e^{-cr} \mathrm{d}r} \tag{17}$$

and then, the number concentration $N$ can be calculated by integrating the Eq. (4). The algorithm flowchart is shown below

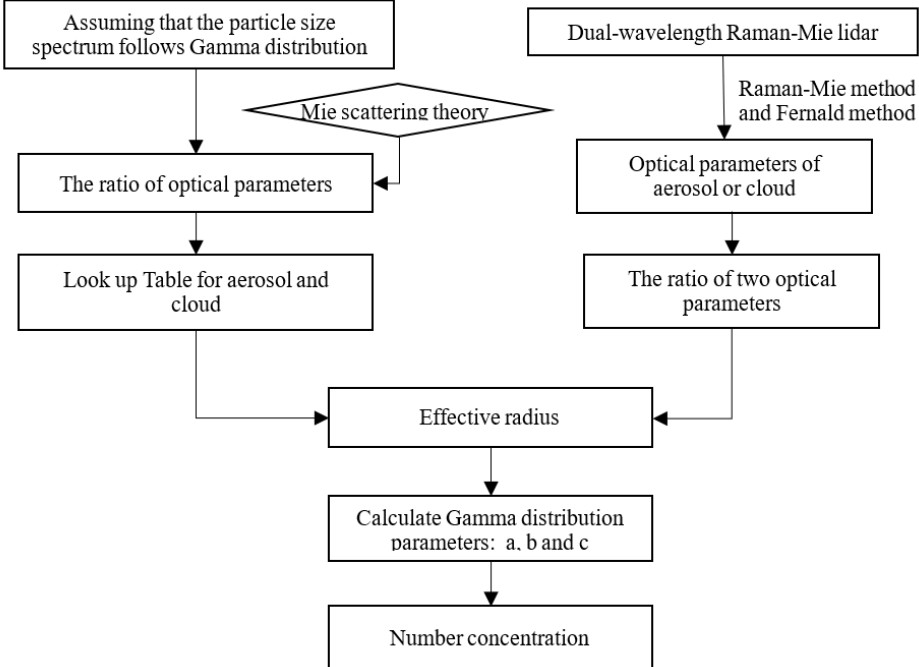

**Figure 2.** The algorithm flowchart for atmosphere particle microphysical parameters.

The algorithm is described as follows:

In this algorithm, the first step is to establish a lookup table between aerosol/cloud optical parameters and microphysical parameters. 1) Assuming that aerosol particles and cloud droplets follow the Gamma distributions, calculate the extinction coefficient and backscatter coefficient at different laser wavelengths (355nm and 1064nm in this paper) based on the Mie scattering theory; 2) Calculate the ratio of backscatter coefficients for two wavelengths, which is the backscatter color ratio, or calculate the ratio of extinction coefficient to backscatter coefficient, which is the radar ratio; 3) Change the parameters of the aerosol to obtain the gamma distributions with effective radius from 0.2 μm to 3 μm, calculate the optical parameters and corresponding optical parameter ratios (radar ratio or backscatter color ratio) for each Gamma distribution, and establish the lookup table for aerosol effective radius; 4)Similar to the step 3, establish the lookup table for cloud drops (effective radius are from 0.5 μm to 5 μm). After the lookup table is completed, the microphysical parameters of aerosols or clouds are calculated based on the lookup tables and LiDAR detection data. The specific steps are as follows:1)the dual-wavelength (355 nm and 1064 nm) Raman LiDAR need be selected for the detection of atmosphere; 2) Raman and Fernald methods are used for the retrieval of optical parameters at multi-wavelengths, and the backscatter color ratio or lidar ratio can be obtained ; 3) aerosol and cloud layers are identified based on lidar echo signals; 4) Retrieve the effective radius of aerosols or cloud droplets at different heights based on optical parameters ratios and lookup tables; 5) Calculate the parameters b and c in the Gamma distribution according to formulas (13) and (16); 6) Calculate the value of a in the Gamma distribution according to the Eq.(17); 7) Calculate the number concentration according to the Eq. (3).

**3.2 The simulation**

**3.2.1 The relationship between lidar ratio, color ratio, and effective radius**

Due to the different complex refractive indices of aerosols and clouds, we will discuss them separately. Water clouds are

composed of liquid droplets, the complex refractive index of $1.33\text{-}10^{-7}i$ was selected. The theoretical relationship curves of
lidar ratio of 355 nm, lidar ratio of 532 nm, 355/1064 nm backscatter color ratio, 355/532 nm backscatter color ratio with
effective radius were calculated and shown in Fig. 3(a) to 3(d).

**Figure 3.** The theoretical relationship curves of colour ratio or lidar ratio with effective radius, $m=1.33-0.10^{-7}i$. (a) Lidar ratio of 355 nm,
(b) lidar ratio of 532 nm, (c) the ratio of backscatter coefficients (355/1064 nm), (d) the ratio of backscatter coefficients (355/532 nm).

The composition of aerosols is complex, with a large variation of complex refractive index, ranging from 1.33 to 1.70 in the
real part and 0 to 0.05 in the imaginary part. Assuming the complex refractive index of aerosols is 1.47-0.002i, Fig. 4(a) to 4(d)
respectively show the theoretical relationship curves of aerosol when parameter $b$ is set to 2-7.

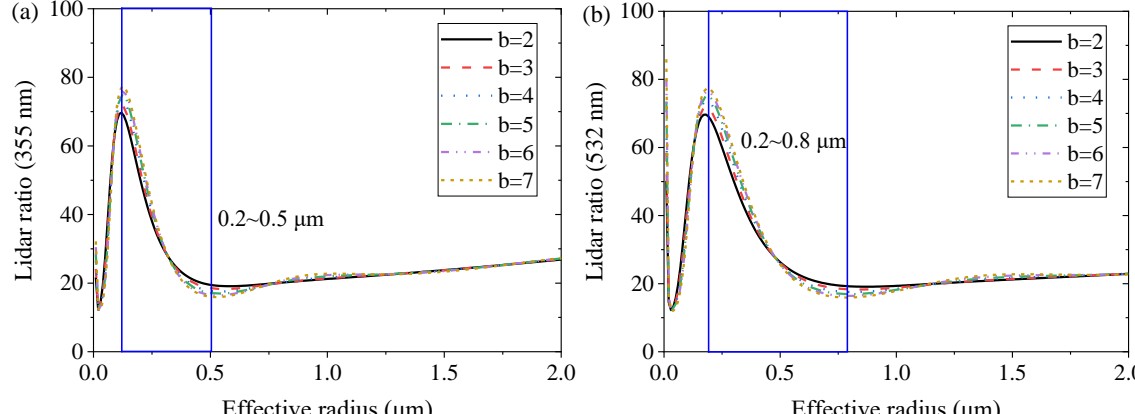

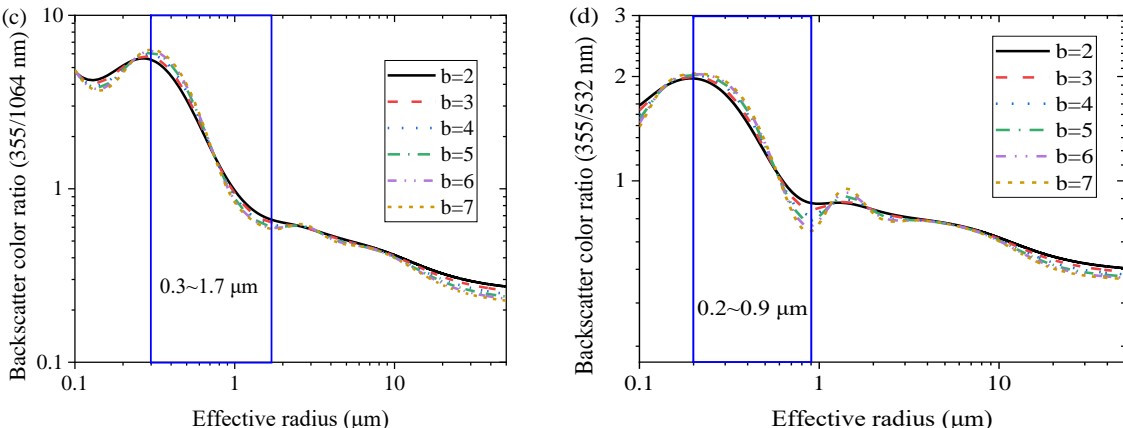


**Figure 4.** The theoretical relationship curves of colour ratio or lidar ratio with effective radius, $m=1.47-0.002i$. (a) Lidar ratio of 355 nm, (b) lidar ratio of 532 nm, (c) the ratio of backscatter coefficients (355/1064 nm), (d) the ratio of backscatter coefficients (355/532 nm).

The blue boxes in Figures 3 and 4 refer to the monotonic variation intervals of aerosols and cloud droplets, respectively. As shown in the figures, when the complex refractive index is constant and the parameter b is set to 2-7 or 2-8, the corresponding curve trend is consistent. Under a constant complex refractive index, parameter b does not change the trend of the curve. The change of *b* has little effect on the curve. If the color ratio (355 nm/1064 nm) is selected for the retrieval of effective radius, the influence of b value on the results is about 5% (as shown in Figure 3c and 4c). If the lidar ratio is selected for the retrieval of effective radius, the influence of b value on the results will be slightly greater, and it might reach ~10% (as shown in Figure 3a and 4a). Within the monotonic interval, the effective radius of particles can be retrieved from the curves. The monotonic interval varies with optical parameter. It can be seen that whether it is clouds or aerosols, the monotonic range of the backscatter color ratio is the widest, as shown in Fig. 3(c) and Fig. 4(c). The larger the value of *b*, the more pronounced the Gamma function describes the characteristics of large particles. Therefore, in the subsequent inversion, *b*=6 is taken for cloud droplets, and *b*=3 for aerosols.

Considering the laser's penetration ability, and the monotonic range of optical parameter ratios with effective radius shown in Fig. 3 and Fig. 4, the backscatter ratio of 355 nm/1064 nm for the inversion is the optimal choice. According to Fig. 3(c), the effective radius that can be retrieved using backscatter ratio of 355 nm/1064 nm is above 1 μm. The optimal inversion range is 1-3.4 μm, and the maximum inversion radius can reach 10 μm. For aerosol particles, the theoretically retrieval effective radius is above 0.3 μm, the optimal inversion interval is 0.3-1.7 μm. The applicability of this algorithm is limited, and it is applicable for aerosols and small cloud droplets. For aerosols, particle diameter is usually 0.01~10 μm, while the effective particle radius ranges from 0.3 μm to 1.2 μm for urban aerosols (This is calculated based on ground and aircraft observation data.). Usually, water droplets diameter larger than 2 microns are called cloud droplets. Therefore, this algorithm is suitable for the detection of urban aerosols and small cloud drops. The above curves in Fig. 3 and Fig. 4 obtained are calculated using Mie scattering theory and are suitable for spherical particles. The spherical particles in the atmosphere can be distinguished from the depolarization ratio.

**3.2.2 The influence of complex refractive index on the backscatter color ratio**

When the complex refractive index changes and *b* is 3, the backscatter color ratios of the 355 nm and 1064 nm wavelengths are shown in Fig. 5(a) to 5(d).

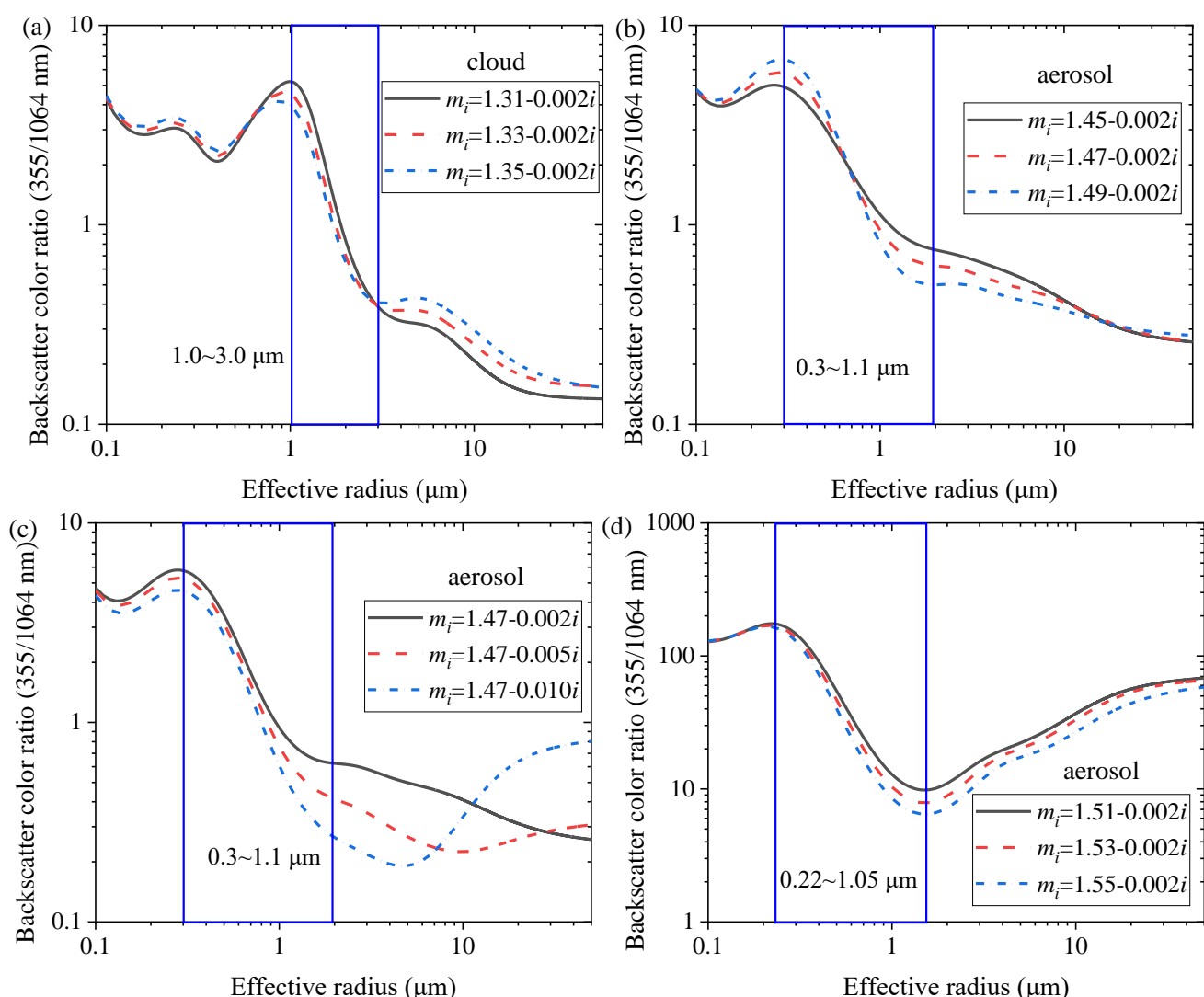

**Figure 5.** The color ratio with different complex refractive indices. (a)Aerosol with different real part of complex refractive index (real part <1.50), (b)aerosol with different imaginary part of complex refractive index (real part <1.50), (c)aerosol with different real part of complex refractive index (real part >1.50), (d)aerosol with different imaginary part of complex refractive index (real part >1.50).

According to Fig. 5, when the complex refractive index of particles changes, the color ratio curves will fluctuate, but they always monotonically decreases from 0.3 μm to 1 μm. Therefore, if the aerosol composition is stable, the color ratio curve can well reflect the trend of effective radius variation.

### 3.2.3 Algorithm verification

The verification of this algorithm is achieved through simulation.The specific steps are as follows: 1) Calculate the effective radius and number concentration using APSD and CDSD observed by aircraft and the equation (8); 2) Calculate the backscatter coefficient at two wavelengths of 355 nm and 1064 nm, and then calculate the color ratio according to the Eq. (13); 3) According to the color ratio and the algorithm described in Figure 2 of Section 3.1, the effective radius and number concentration profiles can be retrieved; 4) Compare the effective radius and numerical concentration in steps 2) and 4), as shown in Fig.6, to verify the algorithm inversion.

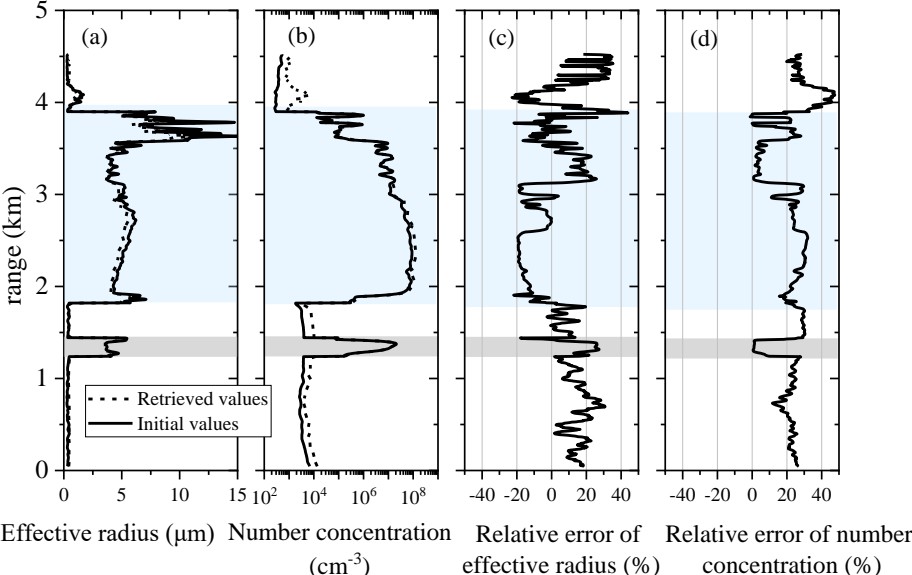

**Figure 6.** Simulation and verification of the algorithm with aircraft data. (a) effective radius, (b) number concentration, (c) effective radius error, (d) number concentration error.

Figure 6(a) and 6(b) are the true values and inversion results of the effective radius and number concentration of an aircraft observation at vertical altitude, respectively. Figure 6(c) and 6(d) show the relative errors, respectively. The light gray and light blue shaded areas in the figures are cloud layers. It can be seen that effective radius and number concentration can be well retrieved using the algorithm. Figure 6 shows that the retrieval error of cloud droplets is relatively small, within ±20% and ±30% for effective radius and number concentration. The errors are ±20% and ±40% for aerosol. The inversion error of microphysical parameters of aerosol particles is larger than that of cloud droplets. The reasons are: 1) aerosol types are more complex, and the assumption of complex refractive index is prone to deviation; 2) APSDs is more complex than CDSDs, and the adaptability to Gamma distribution is relatively low.

### 3.3 Error analysis of the algorithm

The inversion errors of effective radius and number concentration mainly come from three aspects: 1) error introduced by non-spherical particles; 2) error introduced by the assumption of Gamma distribution; 3) error introduced by improper assumption of complex refractive index; 4) error caused by optical parameter inversion deviation.

For urban aerosols and water clouds, their particles are spherical, so the error caused by non-spherical particles can be ignored. The error introduced by the assumption of Gamma distribution is relatively complex and difficult to accurately calculate. This study evaluates this error by numerical simulation based on APSDs and CDSDs data by aircraft observations. Actually, the error presented in Fig. 6 is mainly caused by the assumption of Gamma distribution. Calculate optical parameters of over 5000 sets of APSDs and CDSDs data, and retrieve the microphysical parameters using our algorithm. The calculated standard deviations between the inversion results and the actual data are: for aerosols, the standard deviation of the effective radius is ~10%, and the standard deviation of numerical concentration is ~20%; For clouds, the standard deviation of the effective radius is 15%, and the standard deviation of numerical concentration is ~20%.

The deviation introduced by improper assumption of complex refractive index may be the largest term in this technique. For water clouds, the complex refractive index is stable and the deviation caused by it can be ignored. It is difficult to accurately obtain the complex refractive index of aerosols, and the deviation caused by the complex refractive index may reach over 100%. Figure 6 shows the effect of complex refractive index variation on the optical parameter ratio. From Figure 6, it can be seen that when the real part of the complex refractive index changes within the range of 0.03 and the imaginary part changes within 0.01, the effective radius deviation caused by the complex refractive index is within a controllable range. After calculation, the deviation does not exceed 40%. And it can be seen that although complex refractive index can lead to the significant change of the effective radius value, when the aerosol is constant,

its monotonic characteristics remain unchanged, which means that the evaluation of particle size changes is reliable.
In order to quantitatively analyze the impact of optical parameter errors on the effective radius inversion results, the effective
radius errors caused by color ratio error were calculated when they are ±5% and ±10%, and shown in Fig. 7.

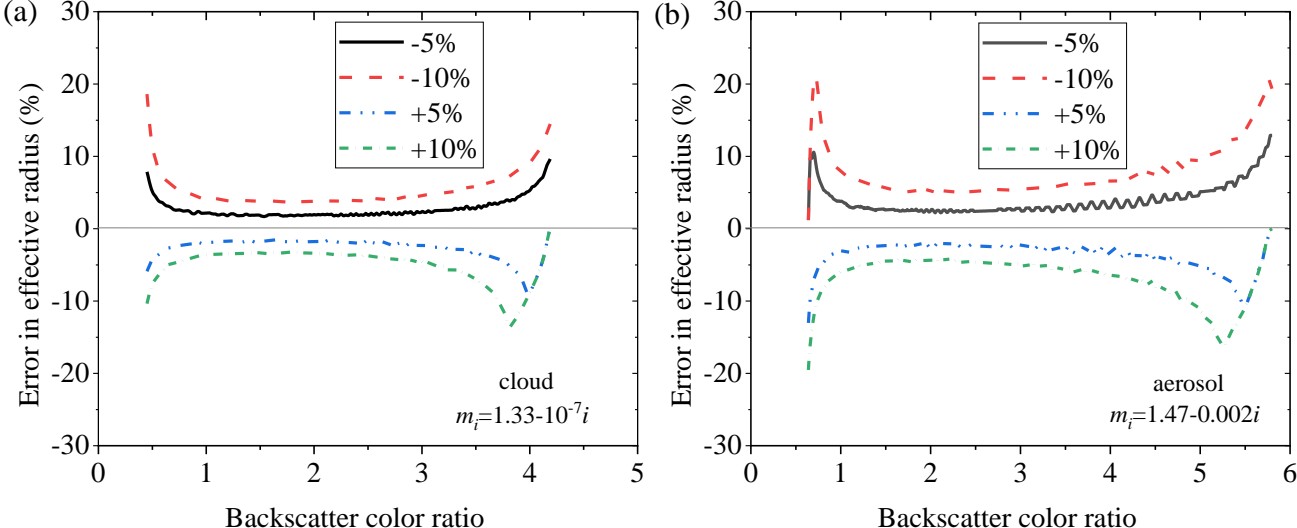


**Figure 7.** Errors in effective radius in Look-Up-Table when there are ±5% and ±10% errors in the backscatter color ratio. (a)Cloud
droplets, (b)aerosol particles.
From Fig. 7 (a), it can be seen that when there are errors of ±5% and ±10% in the backscatter color ratio, the inversion errors
of the effective radius of cloud droplets are within ±10% and ±20%, respectively. According to Fig. 7(b), when there are errors
of ±5% and ±10%, the inversion errors of aerosol effective radius are within ±20% and ±30%, respectively.
Considering the actual inversion ability of LiDAR, the deviation of color ratio will reach 10%. The above errors are
independent of each other. The final evaluation shows that the mean square deviation of the inversion error of aerosol
effective radius is less than 45%, and the standard deviation of the inversion error of cloud droplet effective radius is less
than 25%.
The inversion error of number concentration comes from the final superposition of optical parameter error and effective
radius error, and the error should be slightly larger than effective radius. In this algorithm, the complex refractive index needs
to be assumed. The physical and chemical properties of aerosol particles and cloud droplet particles that interact with the cloud
are similar, with a complex refractive index similar to that of the cloud. Continuous microphysical parameter profiles can be
obtained by this algorithm. For the uniformly mixed aerosol layer, it can be considered that the complex refractive index within
the layer remains unchanged. Therefore, this algorithm is suitable for the inversion of microphysical parameters of uniformly
mixed aerosol particles and small cloud droplet particles.
**4 Experiment**
**4.1 Instrument**
A multi-wavelength (355 nm/532 nm/1064 nm) lidar has been developed in Xi'an University of Technology (XUT). A
Cassegrain telescope is employed as the optical receiver, and narrowband interference filters are utilized as core filter devices
to finely separate the backscatter signals. The system consists of five detection channels: the two elastic scattering channels at
the wavelength of 355 nm and 1064 nm, the nitrogen Raman scattering channel at 387 nm, and the two polarization channels
at 532 nm. Table 1 summarizes the main system parameters of the lidar system.
**Table 1.** System parameters of multi-wavelength Raman-Mie scattering Lidar.

| Instrument | Main instrument parameters | | |
|---|---|---|---|
| Wavelength of laser | 355 nm, 532 nm, 1064 nm | | |
| Light source | Leibao SGR series Nd: YAG pulsed laser | | |

| Multi-wavelength Raman-Mie Scattering Lidar | | Pulse width | 8.4 ns | Repetition frequency | 10 Hz |
|---|---|---|---|---|---|
| | Telescope | Laser divergence angle | | ≤ 0.5 mrad | |
| | | Cassegrain telescopes | | | |
| | | Focal length | 2 m | Field of view | 0.5 mrad |
| | | Aperture | | 400 mm | |
| | Wavelength of signal | 355 nm (Mie channel), 387 nm (Raman channel), 532 nm (Polarization channel), 1064 nm (Mie channel) | | | |
| | Time resolution | 2 min | | | |
| | Distance resolution | 3.75 m | | | |

The optical parameters obtained from this system are the backscatter coefficients at 355 nm ($\beta_{355}$) and 1064 nm ($\beta_{1064}$), extinction coefficient of 355nm ($\alpha_{355}$), depolarization ratio of 532 nm ($\delta_{532}$). $\beta_{355}$ is obtained by inverting the Mie-scattering and Raman channel without assuming lidar ratio. $\beta_{1064}$ can be inverted by the Fernald method, as described in Wang et al (2023a) and Li et al (2016).

**4.2 The experimental observation of cloud layer**

**4.2.1 The experimental observation**

Experimental observations were performed based on the lidar of XUT at the Jinghe National Basic Meteorological Observing Station (34.43°N, 108.97°E) on September 16, 2022 (BJT). The observation experiment lasted for 7 hours with a time resolution of 2 minutes. Figure 8(a) shows the Time Height Intensity (THI) of the Mie-Rayleigh signal at 1064 nm, and the color bar values in the figure are the logarithm of RSCS. Figure 8(b) are the temperature and relative humidity profiles obtained from the sounding balloon at 7:15 am.

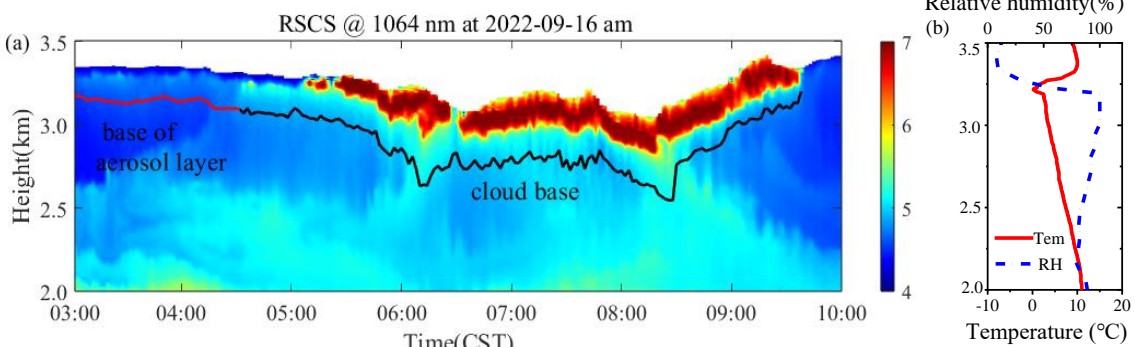

**Figure 8.** Lidar and sounding balloon observations. (a) THI diagram of RSCS at 1064 nm observed by lidar at 03:00-10:00 September 16, 2022(CST), (b) temperature and relative humidity observed by sounding balloon at 07:15 September 16, 2022.

According to Fig. 8(a), there are signals changing from weak to strong above the black curve near 3 km. After 5:00, the echo signal gradually increased and the laser could not penetrate. The red and black lines in the figure correspond to the lower boundaries of the aerosol layer and cloud layer of interest, respectively, calculated by differential zero crossing method. According to the temperature and humidity profiles shown in Fig. 8(b), the temperature below 3.5 km is higher than 0°C, and the relative humidity reaches over 90% at 3 km-3.2 km. Therefore, it can be determined that the strong signal appearing near 3 km in the atmosphere is water cloud.

**4.2.2 The optical and microphysical parameter profiles**

Figure 9 shows the observed signals of the lidar experiment near 5 o'clock on September 16, 2022, as well as the retrieved optical and microphysical parameters. Figure 9(a) is the dual wavelength RSCS with enhanced signal in the cloud, especially at 1064 nm. Figure 9(b) shows the volume depolarization ratio profile. The volume depolarization ratio in aerosols and clouds is less than 0.05, indicating that the detected aerosols and clouds are spherical particles. Figure 9(c) show the dual wavelength backscattering coefficient profiles at 355 nm and 1064 nm, while Fig. 9(d) is the ratio of backscattering coefficients at 355 nm and 1064 nm, i.e., backscatter color ratio (Wang et al., 2023b).

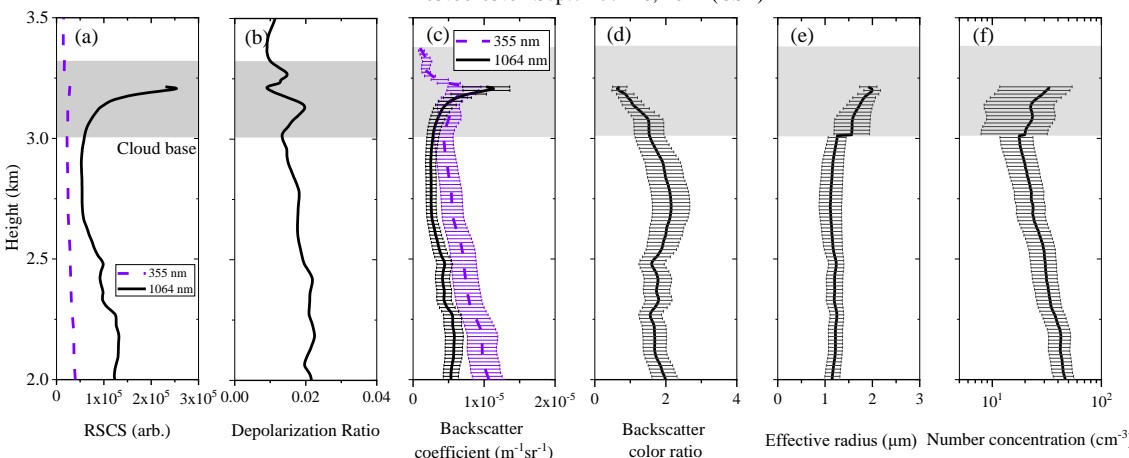

303

**Figure 9.** Lidar observation results at 05:00-05:02 September 16, 2022(CST). (a)Dual-wavelength RSCSs, (b)depolarization ratio, (c)backscatter coefficients, (d)backscatter color ratio, (e)effective radius, (f)number concentration.

The depolarization ratio of aerosols below the cloud layer does not change significantly, indicating that aerosols are uniformly mixed. Based on the inversion algorithm, the effective radius and number concentration profiles are calculated, as shown in Fig. 9(e) and 9(f), respectively. In the process of cloud shown in Figure 8, the aerosol hygroscopicity increase plays an important role. According to Figure 8b, the relative humidity reaches 100% near 3km, and below 3km, the relative humidity is less than 100%. Therefore, the aerosol lookup table is used below the cloud base for the retrieval of aerosol profiles, and the cloud droplet lookup table is used above the cloud base (gray shaded area). The effective radius of aerosols under cloud layer ranges from 1.1 to 1.3 μm, and the concentration fluctuates between 17 and 60 cm$^{-3}$, and the values decrease with increasing height. At the cloud base, the effective radius reaches 1.6 μm, and the concentration is 20 cm$^{-3}$. As the height above the cloud base increases, the effective radius and number concentration both show an increasing trend. The error bars in Fig. 9 represent the uncertainty of the inversion result. The error of backscattering coefficient and backscatter color ratio is determined by the signal-to-noise ratio of the LiDAR system and the error of the optical parameter inversion algorithm. The error bar of the effective radius represents the uncertainty of the results caused by optical parameter errors and gamma distribution assumption errors.

Figure 10 shows the observed signals of the lidar experiment at 7:20 on September 16, 2022, as well as the retrieved optical and microphysical parameters. Compared with Figure 9, RSCS (Fig. 10(a)) and backscatter coefficients (Fig. 10(c)) in the cloud layer increases significantly. From Fig. 10(b), the depolarization ratio increases above 3.2 km, and it should be caused by multiple scattering or low signal-to-noise ratio. The effective radius and numerical concentration of aerosols under the clouds in Fig. 10 show little change compared to Fig. 9. The number concentration in the clouds shown in Fig. 10(f) has significantly increased, reaching ~2000 cm$^{-3}$, but the effective radius didn't change obviously, about 1-2 μm, see Fig. 10(e). According to Fig. 8 (b), it can also be observed that there is a significant inversion layer at 3.2 km, so it is normal for there to be more aerosol accumulation below the inversion layer.

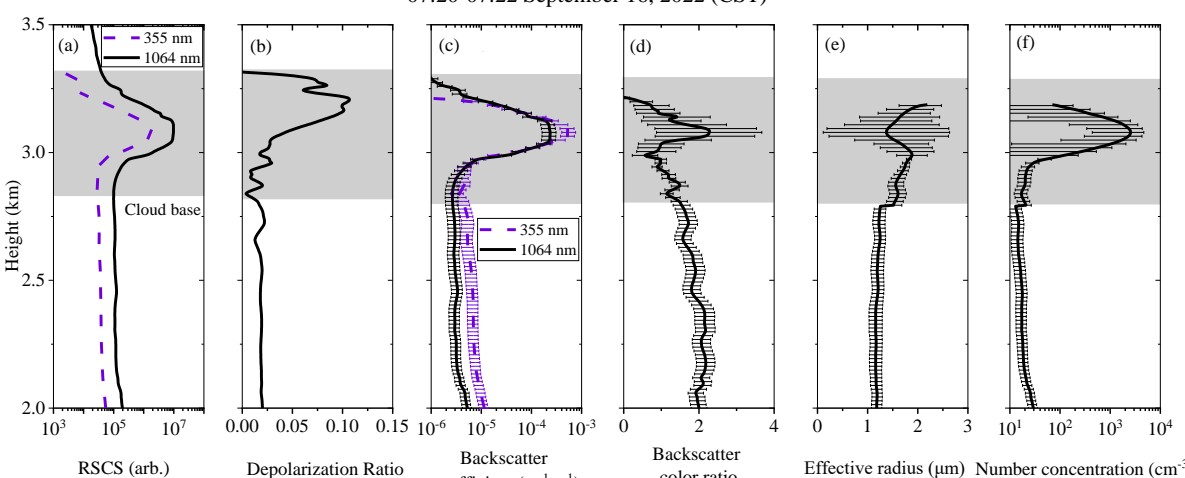

**Figure 10.** Lidar observation results at 07:20-07:22 September 16, 2022(CST). (a)Dual-wavelength RSCSs, (b)depolarization ratio, (c)backscatter coefficients, (d)backscatter color ratio, (e)effective radius, (f)number concentration.

### 4.3 the observation results of cloud process

Figure 11 shows the THI of color ratio. In the region with cloud, the color ratio is relatively small, about 0.5-2, and the color ratio of aerosols is relatively large, about 2-7.

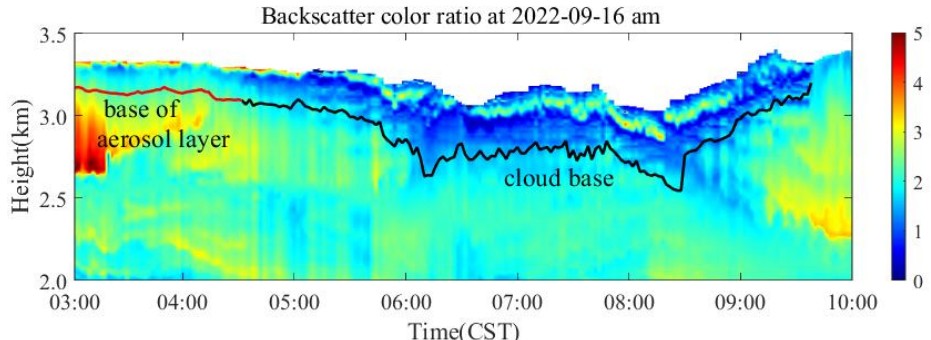

**Figure 11.** Inversion results of backscatter color ratio at 03:00-10:00 September 16, 2022(CST).

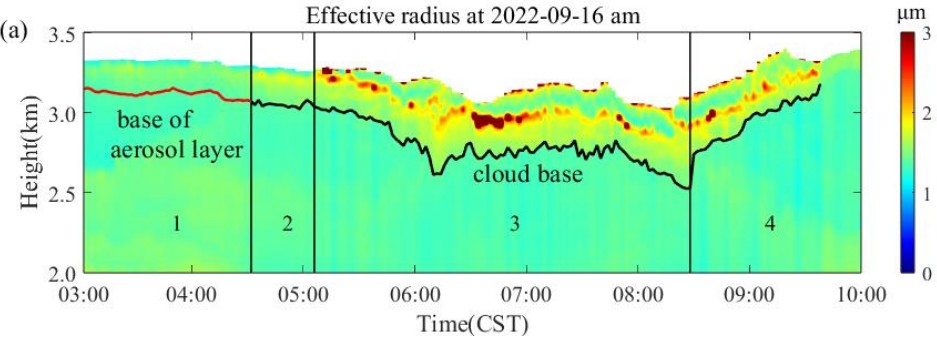

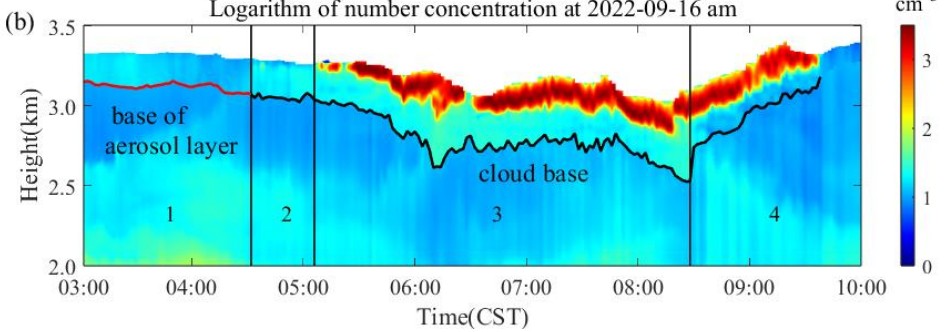

**Figure 12.** Microphysical parameters inversion results of atmospheric particulate matters at 03:00-10:00 September 16, 2022(CST). (a)Effective radius, (b)number concentration.

Figure 12 shows the changes of effective radius and particle number concentration (displayed in logarithmic form). The observation results can be separated into four stages, marked with "1/2/3/4" in Fig. 12. There are no clouds in regions 1 and 2, but based on the lidar echo signal, we can see a more obvious signal growth and change process. Stage 1: From 03:00 to 04:30, clouds have not yet formed, but an obvious layer of aerosol at 3.2 km with an average thickness of 180 m can be seen. The effective radius and number concentration are relatively small, ranging from 1.2 to 1.5 μm and from 8.5 cm$^{-3}$-20.6 cm$^{-3}$. Stage 2: 04:30-05:06, during which the echo signal of the lidar is enhanced, the particles radius increase, and the effective radius increases to 1.4-1.8 μm. The concentration range is 13.3 cm$^{-3}$-25.6 cm$^{-3}$. Stage 3: From 05:00 to 08:30, cloud droplets generated and cloud layer appeared. The echo signal intensifies sharply, and the effective radius and number concentration increase significantly, with the effective radius of 1.5-5.3 μm and the concentration of 18.7 cm$^{-3}$-2853.5 cm$^{-3}$. Due to the increase of number concentration, the laser cannot penetrate the cloud layer. Stage 4: From 08:30 to 10:00, the cloud layer rises and the cloud base height increases from 2.5 km to 3.27 km. The effective radius inside the cloud remains unchanged, but the numerical concentration decreases. At 9:40, the cloud signal disappeared, possibly due to the cloud leaving the field of view of lidar and unable to be observed.

## 5. Conclusion and Discussion

This study proposes a method to estimate the microphysical parameters of atmosphere aerosols and small cloud droplets using two optical parameters. Assuming Gamma distribution, the effective radius and number concentration of aerosols or small cloud droplets can be calculated using the backscatter color ratios of 355nm and 1064nm wavelengths. An atmosphere observation experiment was conducted using the multi-wavelength Lidar, and the effective radius and number concentration were retrieved. The results indicate that the algorithm is stable and reliable.

This algorithm has simple hardware requirements for lidar, requiring only two wavelengths to achieve the retrieval of microphysical parameters. At the same time, the algorithm is simple, and can obtain stable data inversion results. It is suitable for the retrieval of cloud droplet generation process and aerosol with uniform mixing and relatively stable composition. The limitation of this algorithm is that it requires assuming the complex refractive index of particles. The complex refractive index of aerosols varies greatly, and incorrect assumptions about the complex refractive index can have a certain impact on the results. Furthermore, this algorithm is not applicable for retrieval of large particle sizes (radius>10 μm). To detect larger particle sizes, millimeter wave cloud radar and lidar can be used for joint observation. We will carry out this work in the future.

## Data availability

The data and codes related to this article are available upon request from the corresponding author.

## Author contributions

Conceptualization: Huige Di

Investigation: Xinhong Wang & Huige Di

Methodology: Huige Di & Xinhong Wang

Software: Xinhong Wang & Ning Chen

Writing — original draft: Xinhong Wang & Huige Di

Writing — review & editing: Huige Di, Dengxin Hua

Supervision: Huige Di & Jing Guo

Data collation: Wenhui Xin, Shichun Li, Yan Guo, Qing Yan, & Yufeng Wang

Project administration: Huige Di & Dengxin Hua

**Competing interests**
The authors declare that they have no conflicts of interest related to this work.
**Acknowledgements**
We express our gratitude to the Xi'an Meteorology Bureau of Shaanxi Province, Xi'an, Mei Cao for providing the relevant
supporting data.
**Financial support**
This research was supported by the National Natural Science Foundation of China, the Innovative Research Group
Project of the National Natural Science Foundation of China (Grant Nos. 42130612).

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
