# Peer review of "The algorithm of microphysical parameter profiles of aerosol and 1"

_EGUsphere, 2024_

## Community Comment (CC1)

A method for retrieval of aerosol and small cloud droplet microphysical parameters using the backscatter coefficient of two wavelengths (355nm and 1064nm) of lidar is proposed in this manuscript. The algorithm is derived in detail, and the sources of error and applicable conditions of the algorithm are discussed. This algorithm only requires two wavelengths to achieve effective radius and number concentration, and is simple and stable. It is suitable for inversion of cloud base cloud droplet and aerosol with uniform mixing and relatively stable composition. The method proposed is innovative and of practical value. However, this algorithm also has certain limitations, namely the scale of particles that can be inverted is limited.

**Response: Thank you very much for your nice comments. Your question and suggestion are very helpful for us to improve the quality of our paper. We appreciate the reviewer's thoughtful review and constructive comments. The following is our point-to-point replies.**

Specific Comments:

1. How to determine whether the detected object meets the scope of application of the algorithm?

**Answer: This algorithm is applicable for aerosols and small cloud droplets. For aerosols, particle diameter is usually 0.01~10um, while the effective particle diameter ranges from 0.6um to 1.2um for urban aerosols. Usually, water droplets larger than 2 microns are called cloud droplets. Therefore, this algorithm is suitable for the detection of urban aerosols and initial cloud formation, or for cloud lateral boundary.**

2. The aerosol size distribution and cloud droplet size distribution used in section 2.2 were obtained in 2005 to 2006, and if new statistical data can be used, the conclusion would be more convincing.

**Answer: Thanks, the reviewer is absolutely right. However, aircraft observation data is rare and precious, and this is the only data we can obtain. This data is also representative.**

3. What is the impact of b-value changes in the Gamma distribution on the results? Quantitative data needs to be provided in the manuscript.

**Answer: According to Figures 4 and 5, we can also observe that the influence of b value on the results is not significant.**

**The influence of b value on the results varies depending on the parameters selected. If the color ratio (355nm/1064nm) is selected for the retrieval of effective radius, the influence of b value on the results is about 5% (as shown in Figure 3c). If the lidar ratio is selected for the retrieval of effective radius, the influence of b value on the results will be slightly greater, and it might reach ~10% (as shown in Figure 3c).**

4. In Figure 10, there is a sharp increase in the echo signal above the cloud layer. Is this caused by ice crystals?

**Answer: According to the temperature profile in Figure 8b, it can be seen that**

temperatures below 3.5 km are generally greater than 0 ˚C. Therefore, it can be concluded that the strong signal on the cloud layer is not caused by ice crystals. According to Figure 12b, it can be seen that the large values are caused by the increase of particles number.

5. Table1, Resolvable time? Minimum resolvable distance?

Answer:"Resolvable time" should be "time resolution".

"Minimum resolvable distance" should be "distance resolution".

6. There are grammatical errors in the manuscript, which need to be carefully revised.

Answer:We will check and revise the manuscript sentence by sentence to avoid mistakes. Thanks.

---

## Community Comment (CC2)

This study proposed an inversion method of atmosphere aerosol or cloud microphysical parameters based on dual wavelength lidar data. However, several comments below should be settled.

**Response: Thank you very much for your nice comments. Your question and suggestion are very helpful for us to improve the quality of our paper. We appreciate the reviewer's thoughtful review and constructive comments. The following is our point-to-point replies.**

1. Line 31: an5bd?

**Answer: it is a clerical error. It should be "and". We will check and revise the manuscript sentence by sentence to avoid mistakes. Thanks.**

2. Part 2.2: This section needs to be improved by expanding to include more supportive figures and detailed descriptions.

**Answer: Thanks for your nice comments.**

This section has been modified as

"In order to study the characteristics of APSDs and CDSDs in the vertical altitude, the APSDs and CDSDs obtained from aircraft observations by the Hebei Provincial Weather Modification Office were analyzed (from 2005 to 2006). APSDs were measured by the PCASP-100X probe, and CDSDs were obtained by the FSSP-100-ER probe [29]. The PCASP-100X is an optical particle counter for measuring aerosol size distribution from 0.10 μm to 3.00 μm in diameter in 15 different size bins with a frequency of 1 Hz. The sample flow volume in the PCASP-100X was set to 1 cm$^3$ s$^{-1}$. FSSP-100-ER is an instrument that measure cloud droplet size and concentration using light scattering, with the measurement range of 0.5-47 μm.

The obtained APSDs and CDSDs were fitted one by one using Gamma function. In order to minimize the error at all radius, the minimization problem is solved using the following equation

$$\int_0^{D_{max}} \left( \log\left(f_m(D)\right) - \log\left(f_{fitted}(D)\right) \right)^2 dD \rightarrow \min \tag{12}$$

here, $f_m(D)$ is the actual particle size distribution measured by the PCASP-100X, $f_{fitted}(D)$ is the fitted distribution, $D$ is the aerosol particle diameter, $D_{max}$ is the measured maximum particle diameter. The goodness of fit $R^2$ is used to represent the difference between the fitting function and the measured data. The definition of goodness of fit is as follows::

$$R^2 = 1 - \frac{\sum_{i=1}^{n}\left(y_i - \hat{y}_l\right)^2}{\sum_{i=1}^{n}\left(y_i - \overline{y}\right)^2} \tag{13}$$

where $y_i$ is the measured value, $\hat{y}_l$ is the predictive value, $\overline{y}$ is the mean measured value. The numerator represents the sum of squared residuals, and the denominator represents the sum of squared total deviations.

~3500 sets of APSDs and 2221 sets of CDSDs were statistically analyzed. Over 95% of the data have a high goodness of fit in the Gamma distribution. The goodness of fit of CDSDs is higher than that of APSDs, with CDSDs of 0.983 and APSDs of 0.856. The parameters of CDSDs are significantly larger than that of APSDs, and there are obvious differences of b and c for cloud and

aerosol. The literature suggests that there is a certain functional relationship between the Gamma parameters b and c of CDSDs [30]. Statistical analysis was conducted on the b and c parameters of APSDs and CDSDs, as shown in Fig. 1.

[Figure]

Fig. 1 Statistical Results of parameter $b$ and $c$ in aerial survey data. (a)Aerosol particles, (b)cloud droplets.

According to Fig. 2, there are the remarkable linear relationships between parameter b and c. The fitting functions for CDSDs and APSDs are as follows:

$$\begin{cases} c_{\text{cloud}} = 0.33 b_{\text{cloud}} + 0.60 \\ c_{\text{aerosol}} = 2.67 b_{\text{aerosol}} + 7.43 \end{cases} \tag{14}$$

The linear relationship between the two parameters of CDSDs is better with a goodness of fit of 0.948, and a linear goodness of fit of 0.821 for APSDs. According to the statistical results, the parameter b of APSDs at vertical height is mainly distributed in the range of 2-7, and CDSDs is mainly distributed in the range of 2-8. "

3. Figure 2: Not clear, Add more details on the description of the algorithm. For example, the look-up-table, etc.

**Answer: Thanks for your nice comments.**

**We have redrawed the flowchart of the algorithm, and shown as the following figure.**

[Figure]

Figure 1 the algorithm flowchart for atmosphere particle microphysical parameters

And the descriptions of the algorithm are added.

"In this algorithm, the first step is to establish a lookup table between aerosol/cloud optical parameters and microphysical parameters. 1) Assuming that aerosol particles and cloud droplets follow the Gamma distributions, calculate the extinction coefficient and backscatter coefficient at different laser wavelengths (355nm and 1064nm in this paper) based on the Mie scattering theory; 2) Calculate the ratio of backscatter coefficients for two wavelengths, which is the backscatter color ratio, or calculate the ratio of extinction coefficient to backscatter coefficient, which is the radar ratio; 3) Change the parameters of the aerosol to obtain the gamma distributions with effective radius from 0.2 μm to 3 μm, calculate the optical parameters and corresponding optical parameter ratios (radar ratio or backscatter color ratio) for each Gamma distribution, and establish the lookup table for aerosol effective radius; 4) Similar to the step 3, establish the lookup table for cloud drops (effective radius are from 0.5 μm to 5 μm). After the lookup table is completed, the microphysical parameters of aerosols or clouds are calculated based on the lookup tables and LiDAR detection data. The specific steps are as follows:1)the dual-wavelength (355 nm and 1064 nm) Raman LiDAR need be selected for the detection of atmosphere; 2) Raman and Fernald methods are used for the retrieval of optical parameters at multi-wavelengths, and the backscatter color ratio or lidar ratio can be obtained ; 3) aerosol and cloud layers are identified based on lidar echo signals; 4) Retrieve the effective radius of aerosols or cloud droplets at different heights based on optical parameters ratios and lookup tables; 5) Calculate the parameters b and c in the Gamma distribution according to formulas (11) and (14); 6) Calculate the value of a in the Gamma distribution according to formula 18; 7) Calculate the number concentration according to formulas (2) and (5). "

4. line 136: How to determine the boundary of the blue box?

**Answer: The boundary of the blue box is determined by the common monotonic variation interval of multiple curves. Thanks.**

5. line 143: you claimed that the larger the value of b, the more pronounced the Gamma function describes the characteristics of large particles. Why did you choose b=6 for cloud droplets, and b=3 for aerosols?

**Answer:Because the particle size of cloud droplets is larger than that of aerosols, it is preferred to choose the b-value of cloud droplet as the larger value and the aerosol as the smaller value. On the other hand, according to Figures 4 and 5, we can also observe that the influence of b value on the results is not significant.**

6. Figure 5: these figures do not match the description. The 3.2.2 part should be modified.

**Answer:"when the complex refractive index of particles changes, the color ratio curves will fluctuate, but they always monotonically decreases at 0.3 μm to 1.7 μm." has been modified to "when the complex refractive index of particles changes, the color ratio curves will fluctuate, but they always monotonically decreases at 0.3 μm to 1 μm." Thanks.**

7. Line 159-160: It is claimed that "According to Fig. 5, when the complex refractive index of particles changes, the color ratio curves will fluctuate, but they always monotonically decrease at 0.3 μm to 1.7 μm.". However, the content displayed in some figures does not align with the aforementioned description. Furthermore, why is there a need to emphasize "0.3 μm to 1.7 μm"? How do the authors determine these two boundary values?

**Answer:This question is the same as the previous one. There is an error in the manuscript,and it should be "they always monotonically decrease at 0.3 μm to 1 μm." this error will be modified in the revised manuscript. Thanks.**

8. Figure 6: The text of the legend of Figure6(a) and 6(b) such as Inversion value and True value should be modified.

**Answer:Thanks**

**"True value" has been modified to "the initial values" , and "Inversion value" has been modified to "the retrieved values". And the Figure 6 are shown as:**

[Figure]

Figure 6

9. Part 3.3, line 182-183: It is claimed that the first three factors have been discussed earlier and this section focuses on the inversion error introduced by optical parameters. As the error analysis of the algorithm, the all factors which affected the inversion algorithm should be discussed.

**Answer:Yes, the opinion of reviewer is right that "all factors which affected the inversion algorithm should be discussed". More discussions and explanations about error analysis of the algorithm will be added to the revised draft.**

**For urban aerosols and water clouds, their particles are spherical, so the error caused by non-spherical particles can be ignored.**

**The error introduced by the assumption of Gamm distribution is relatively complex and difficult to accurately calculate. This study evaluates this error by numerical simulation based on APSDs and CDSDs data by aircraft observations. Actually, the error presented in Figure 7 is mainly caused by the assumption of Gamm distribution. Calculate optical parameters of over 5000 sets of APSDs and CDSDs data, and retrieve the microphysical parameters using our algorithm. The calculated standard deviations between the inversion results and the actual data are: for aerosols, the standard deviation of the effective radius is ~10%, and the standard deviation of numerical concentration is 20%; For clouds, the standard deviation of the effective radius is 15%, and the standard deviation of numerical concentration is 20%.**

**The deviation introduced by improper assumption of complex refractive index may be the largest term in this technique. For water clouds, the complex refractive index is stable and the deviation caused by it can be ignored. It is difficult to accurately obtain the complex refractive index of aerosols, and the deviation caused by the complex refractive index may reach over 100%. Figure 6 shows the effect of complex refractive**

index variation on the optical parameter ratio. From Figure 6, it can be seen that when the real part of the complex refractive index changes within the range of 0.03 and the imaginary part changes within 0.01, the effective radius deviation caused by the complex refractive index is within a controllable range. After calculation, the deviation does not exceed 40%. And it can be seen that although complex refractive index can lead to the significant change of the effective radius value, when the aerosol is constant, its monotonic characteristics remain unchanged, which means that the evaluation of particle size changes is reliable.

The above three errors are independent of each other. Considering the actual inversion ability of LiDAR, the deviation of color ratio will reach 10%. The final evaluation shows that the mean square deviation of the inversion error of aerosol effective radius is less than 45%, and the standard deviation of the inversion error of cloud droplet effective radius is 25%.

10. Part 4.2: the experimental observation of a cloud generation process was provided in this part. How can the experiment be confirmed as a test of the cloud generation process, rather than clouds drifting in from other locations?

**Answer:**
**When using LiDAR for vertical detection at fixed points, it often detects clouds floating over from other places. However, we believe that Figure 9 shows a process of cloud generation, as we can see from Figure 9 that the aerosol gradually thickens and the cloud gradually thickens, so it should be the generation process of cloud. Clouds drifting from other places usually do not have such clear signal growth process, and the boundaries are usually very clear.**

11. Figure 8: The description of the Figure 8 should be modified. Figure 8(b) is not the lidar observations.

**Answer: Thanks,Figure 8(b) is not the lidar observations,and it is the temperature and relative humidity profiles obtained from the sounding balloon at 7:15 am (BJT).**

12. Figure 9 and Figure 10: What the error bars stand for?

**Answer: The error bar represents the uncertainty of the inversion result. The error of backscattering coefficient and backscatter color ratio is determined by the signal-to-noise ratio of the LiDAR system and the error of the optical parameter inversion algorithm. The error bar of the effective radius represents the uncertainty of the results caused by optical parameter errors and gamma distribution assumption errors.**

13. Only one experimental observation was provided. Could you provide more experiments?

**Answer: Our experimental observation data has other cases, as shown in the following figures, which is another case of observed cloud formation process. The variation process of effective radius and numerical concentration can be clearly displayed. However, this article focuses on the research of data inversion methods, so one case in**

[Figure]

Fig. I Lidar observations at 20:00 June 7 to 4:00 June 8, 2022(CST) (THI diagram of RSCS at 1064 nm)

[Figure]

[Figure]

Fig. II Microphysical parameters inversion results of atmospheric particulate matters at 20:00 June 07-04:00 June 08, 2021(CST). (a)Effective radius, (b)number concentration

---

## Author Response (AR1)

**Response to Editor**

**Dear Editor & Prof.**

We greatly thank you and the reviewers for the thorough and valuable suggestions to our work, and thank the valuable comments and suggestions by peer experts in the open discussion. We have made a point-to-point response to these opinions and suggestions, and all the comments have been addressed in the revised manuscript. We believe that the quality of the manuscript has been promoted now. The responses to each comment are given below.

Thank you very much for considering our work!

Yours sincerely,

Huige Di
Xi'an University of Technology
dihuige@xaut.edu.cn

**Response to Anonymous Referee #1**

A method for retrieval of aerosol and small cloud droplet microphysical parameters using the backscatter coefficient of two wavelengths (355nm and 1064nm) of lidar is proposed in this manuscript. The algorithm is derived in detail, and the sources of error and applicable conditions of the algorithm are discussed. This algorithm only requires two wavelengths to achieve effective radius and number concentration, and is simple and stable. It is suitable for inversion of cloud base cloud droplet and aerosol with uniform mixing and relatively stable composition. The method proposed is innovative and of practical value. However, this algorithm also has certain limitations, namely the scale of particles that can be inverted is limited.

**Response:** Thank you very much for your nice comments. Your question and suggestion are very helpful for us to improve the quality of our paper. We appreciate the reviewer's thoughtful review and constructive comments. The following is our point-to-point replies.

Specific Comments:

1. How to determine whether the detected object meets the scope of application of the algorithm?

**Answer:** This algorithm is applicable for aerosols and small cloud droplets. For aerosols, particle diameter is usually 0.01~10um, while the effective particle diameter ranges from 0.6um to 1.2um for urban aerosols. Usually, water droplets larger than 2 microns are called cloud droplets. Therefore, this algorithm is suitable for the detection of urban aerosols and initial cloud formation, or for cloud lateral boundary.

2. The aerosol size distribution and cloud droplet size distribution used in section 2.2 were obtained in 2005 to 2006, and if new statistical data can be used, the conclusion would be more convincing.

**Answer:** Thanks, the reviewer is absolutely right. However, aircraft observation data is rare and precious, and this is the only data we can obtain. This data is also representative.

3. What is the impact of b-value changes in the Gamma distribution on the results? Quantitative data needs to be provided in the manuscript.

**Answer:** According to Figures 3 and 4, we can also observe that the influence of b value on the results is not significant.

The influence of b value on the results varies depending on the parameters selected. If the color ratio (355nm/1064nm) is selected for the retrieval of effective radius, the influence of b value on the results is about 5% (as shown in Figure 3c and 4c). If the lidar ratio is selected for the retrieval of effective radius, the influence of b value on the results will be slightly greater, and it might reach ~10% (as shown in Figure 3a and 4a).

4. In Figure 10, there is a sharp increase in the echo signal above the cloud layer. Is this caused by ice crystals?

**Answer:** According to the temperature profile in Figure 8b, it can be seen that

temperatures below 3.5 km are generally greater than 0 ˚C. Therefore, it can be concluded that the strong signal on the cloud layer is not caused by ice crystals. According to Figure 12b, it can be seen that the large values are caused by the increase of particles number.

5. Table1,Resolvable time? Minimum resolvable distance?

**Answer:** "Resolvable time" should be "time resolution".

"Minimum resolvable distance" should be "distance resolution".

6. There are grammatical errors in the manuscript, which need to be carefully revised.

**Answer:** We will check and revise the manuscript sentence by sentence to avoid mistakes. Thanks.

**Response to Anonymous Referee #2**

The manuscript presents a method to derive microphysical observations from lidar observations at different wavelengths. Such lidar-microphysical retrieval schemes are of great importance in studying aerosol and clouds. However, I have major concerns regarding the feasibility of the technique given the systematic effects on the retrieval of the particle backscattering coefficient (needed for your approach), which is known to be a difficult retrievable for cloudy situations, due to the lack of a reference (aerosol-free) height to calibrate the lidar. It is also difficult to see a practical use of the method, considering the limited retrievable range of sizes of the droplets (or aerosols). Furthermore, to my opinion, the different sections of the study were not developed and discussed deeply enough. Major revisions need to be made.

**Response:** Thank you very much for your nice comments. Your question and suggestion are very helpful for us to improve the quality of our paper. We appreciate the reviewer's thoughtful review and constructive comments. The following is our point-to-point replies.

Please see my specific comments below:

Page 1 (Introduction): You introduced the problem of getting microphysical information about aerosol and clouds, but as aerosol particles and cloud hydrometeors have been historically approached in different ways, you need to introduce the approaches separately. On the one hand, spectrally resolved information has shown a potential to retrieve microphysical information in the case of aerosol particles, in the case of water clouds, there are quite some limitations because of the larger sizes compared to aerosols.

For this reason, in the case of clouds, there have been several studies that have tried to get information using lidar/radar synergy or lidar-only approaches based on multiple scattering that can be evaluated using dual- or multiple-FOV lidar. A thoughtful literature review of current cloud-retrieval techniques is missing in the manuscript.

**Answer:** Thank you for your nice comments. A thoughtful literature review of current cloud-retrieval techniques has been added in the revised manuscript, and they are " For clouds, there are two methods used for the detection of cloud microphysical parameters. The first method is using lidar/radar synergy for cloud microphysical parameters [a1-a3], which can achieve the retrieval of cloud droplet with large cloud particles. For thin and sparse clouds or nascent clouds, cloud droplet particles are usually small and cannot be detected by millimeter wave cloud radar, which affects the application of this method. The second method is to use multiple scattering information in clouds detected by multi field of view (FOV) or dual FOV LiDAR to retrieval microphysical parameters of water clouds [b1]. However, in order to obtain multiple scattering signals using ground-based lidar, the larger FOV of telescope is required, which will greatly affect daytime detection. "

a1. Zhien Wang and Kenneth Sassen. Cirrus Cloud Microphysical Property Retrieval Using Lidar and Radar Measurements. Part I: Algorithm Description and Comparison with In Situ Data. Journal of applied Meteorology.

2002, 41: 218-229.

a2. Jothiram Vivekandan, Virendra P. Ghate, Jorgen B. Jensen, Scott M. Ellis, and M. Christian Schwartz. ATechnique for Estimating Liquid Droplet Diameter and Liquid Water Content in Stratocumulus Clouds Using Radar and Lidar Measurements. Journal of Atmospheric and Oceanic Technology, 2020, 37: 2145-2161.

a3. Yinchao Zhang, Su Chen, Wangshu Tan, Siying Chen, He Chen, Pan Guo, Zhuoran Sun, Rui Hu, Qingyue Xu, Mengwei Zhang, Wei Hao and Zhichao Bu. Retrieval of Water Cloud Optical and Microphysical Properties from Combined Multiwavelength Lidar and Radar Data. Remote Sens. 2021, 13, 4396.

b1. Nanchao Wang, Kai Zhang, Xue Shen, Yuan Wang, Jing Li, Chengcai Li, Jietai Mao, Aleksey Malinkad, Chuanfeng Zhao, Lynn M.Russellf, Jianping Guo, Silke Gross, Chong Liu, Jing Yang, Feitong Chen, Lingyun Wu, Sijie Chen, Ju Ke, Da Xiao, Yudi Zhou , Jing Fang, and Dong Liu. Dual-field-of-view high-spectral-resolution lidar: Simultaneous profiling of aerosol and water cloud to study aerosol–cloud interaction. PNAS, 2022, 119(10): e2110756119

Eq. 2: It is not quite clear how one gets this equality using the gamma function. Can you add some more steps and explanations to this derivation? And, how valid is it to use the gamma function definition in an integration that does not go up to infinity?

**Answer:** Eq.(3) in the manuscript is the definition of the Gamma function in the positive real number field in mathematics. When applied to aerosol particles or cloud droplets, although the radius is arbitrary, the particle radius interval cannot be infinitely small. The number of particles within each radius bin is also an integer. We use functions to fit the particle number concentration distribution, such as Junge Distribution, Modified Gamma Distribution, Lognormal Distribution, etc., in order to seek more intuitive ways to describe the particle number concentration distribution. Simplifying the distribution of discrete multi-channel particle numbers with radius to a Gamma distribution containing only three parameters may indeed reduce the accuracy of describing the number concentration spectrum, but it is beneficial for understanding the average characteristics and spatiotemporal distribution of particles.

In the revised manuscript, we have swapped the order of formulas (3) and (4).

Line 82: Is there a reference to cite for the cloud probe? FSSP-100-ER. You only include a reference for the aerosol probe.

**Answer :** FSSP-100-ER is an instrument that measures cloud droplet size and concentration using light scattering, with the measurement range of 0.5-47 μm. This has been added in the revised manuscript.

Line 79: These are quite interesting results. Could you deepen the meaning of those parameters? In principle, the b parameter is related to the width and c is related to the size. Is Figure 1 saying, there is always a linear relationship between the size and the width of the distribution?

**Answer:** In principle, the b parameter is related to the width and c is a slope parameter. Figure 2.1 shows the statistical relationship between b and c, which was obtained from the statistical results between the aerosol size distribution and cloud droplet size distribution observed by the aircraft.

Line 110-11: How exactly can you derive the effective radius from Eq. 14?

**Answer:** The detailed descriptions of the algorithm will be added in the revised manuscript.

"In this algorithm, the first step is to establish a lookup table between aerosol/cloud optical parameters and microphysical parameters. 1) Assuming that aerosol particles and cloud droplets follow the Gamma distributions, calculate the extinction coefficient and backscatter coefficient at different laser wavelengths (355nm and 1064nm in this paper) based on the Mie scattering theory; 2) Calculate the ratio of backscatter coefficients for two wavelengths, which is the backscatter color ratio, or calculate the ratio of extinction coefficient to backscatter coefficient, which is the radar ratio; 3) Change the parameters of the aerosol to obtain the gamma distributions with effective radius from 0.2 µm to 3 µm, calculate the optical parameters and corresponding optical parameter ratios (radar ratio or backscatter color ratio) for each Gamma distribution, and establish the lookup table for aerosol effective radius; 4)Similar to the step 3, establish the lookup table for cloud drops (effective radius are from 0.5 µm to 5 µm). After the lookup table is completed, the microphysical parameters of aerosols or clouds are calculated based on the lookup tables and LiDAR detection data. The specific steps are as follows:1)the dual-wavelength (355 nm and 1064 nm) Raman LiDAR need be selected for the detection of atmosphere; 2) Raman and Fernald methods are used for the retrieval of optical parameters at multi-wavelengths, and the backscatter color ratio or lidar ratio can be obtained ; 3) aerosol and cloud layers are identified based on lidar echo signals; 4) Retrieve the effective radius of aerosols or cloud droplets at different heights based on optical parameters ratios and lookup tables; 5) Calculate the parameters b and c in the Gamma distribution according to formulas (13) and (16); 6) Calculate the value of a in the Gamma distribution according to formula (17); 7) Calculate the number concentration according to formula (3). "

Fig 3, Fig 4: It is not stated how the size range limits were defined (the blue lines). How can one assume one is on this range only using, e.g., the backscattering ratio 355/1064? Smaller particle sizes ( left side of the range) might also produce similar ratio values. There is no uniqueness in the parameter you propose to use.

**Answer:** The applicability of this algorithm is limited, and it is applicable for aerosols and small cloud droplets. For aerosols, particle diameter is usually 0.01~10 µm, while the effective particle diameter ranges from 0.3 µm to 1.2 µm for urban aerosols (This is calculated based on ground and aircraft observation data.). Usually, water droplets larger than 2 microns are called cloud droplets. Therefore, this algorithm is suitable for the detection of urban aerosols and initial cloud formation, or for cloud lateral boundary.

Line 164: How do you exactly verify the algorithms? How were the backscatter and lidar ratios calculated from the size distributions? Please provide more accurate information on what is obtained from the measurements.

**Answer**:Thanks for your nice comments.

   The verification of this algorithm is achieved through simulation. The specific steps are as follows: 1) Calculate the effective radius and number concentration using APSD and CDSD observed by aircraft and formula (9); 2) Calculate the backscatter coefficient at two wavelengths of 355 nm and 1064 nm according to the formula (1a), and then calculate the color ratio; 3) According to the color ratio and the algorithm described in Figure 3, the effective radius and number concentration profiles can be retrieved; 4) Compare the effective radius and numerical concentration in steps 2) and 4), as shown in Figure 6, to verify the algorithm inversion.

   According to formula (1a), the backscattering coefficient can be calculated from the APSD or CDSD.

$$\alpha(\lambda) = \int_{r_{\min}}^{r_{\max}} \pi r^2 Q_{\text{ext}}(m, r, \lambda) n(r) \mathrm{d}r \tag{1a}$$

$$\beta(\lambda) = \int_{r_{\min}}^{r_{\max}} \pi r^2 Q_{\text{b}}(m, r, \lambda) n(r) \mathrm{d}r \tag{1b}$$

here, $\alpha(\lambda)$ and $\beta(\lambda)$ are the extinction coefficient and backscattering coefficient at wavelength of $\lambda$, $m$ is the complex refractive index, $Q_{\text{ext}}(m, r, \lambda)$ and $Q_{\text{b}}(m, r, \lambda)$ are the extinction and backscattering efficient factor, $n(r)$ is the number particle size distribution. Lidar ration can be calculated from the following equation.

$$\text{LR} = \frac{\alpha(\lambda)}{\beta(\lambda)} \tag{1c}$$

Line 179: Only theoretical errors are considered, what about systematic errors, such as in the retrieval of the backscattering coefficient, first needed to initialize the retrieval of microphysical properties?

**Answer**:More discussions and explanations about error analysis of the algorithm has be added to the revised draft.

   For urban aerosols and water clouds, their particles are spherical, so the error caused by non-spherical particles can be ignored.

   The error introduced by the assumption of Gamm distribution is relatively complex and difficult to accurately calculate. This study evaluates this error by numerical simulation based on APSDs and CDSDs data by aircraft observations. Actually, the error presented in Figure 7 is mainly caused by the assumption of Gamm distribution. Calculate optical parameters of over 5000 sets of APSDs and CDSDs data, and retrieve the microphysical parameters using our algorithm. The calculated standard deviations between the inversion results and the actual data are: for aerosols, the standard deviation of the effective radius is ~10%, and the standard deviation of numerical concentration is 20%; For clouds, the standard deviation of the effective radius is 15%, and the standard deviation of numerical concentration is 20%.

   The deviation introduced by improper assumption of complex refractive index may be the largest term in this technique. For water clouds, the complex refractive index is stable and the deviation caused by it can be ignored. It is difficult to accurately obtain the

complex refractive index of aerosols, and the deviation caused by the complex refractive index may reach over 100%. Figure 6 shows the effect of complex refractive index variation on the optical parameter ratio. From Figure 6, it can be seen that when the real part of the complex refractive index changes within the range of 0.03 and the imaginary part changes within 0.01, the effective radius deviation caused by the complex refractive index is within a controllable range. After calculation, the deviation does not exceed 40%. And it can be seen that although complex refractive index can lead to the significant change of the effective radius value, when the aerosol is constant, its monotonic characteristics remain unchanged, which means that the evaluation of particle size changes is reliable.

The above three errors are independent of each other. Considering the actual inversion ability of LiDAR, the deviation of color ratio will reach 10%. The final evaluation shows that the mean square deviation of the inversion error of aerosol effective radius is less than 45%, and the standard deviation of the inversion error of cloud droplet effective radius is 25%.

Line 222: how is the black curve defined/determined?

**Answer:** The authors have made modifications and added explanations for the boundary of the aerosol layer and the cloud layer in the original text's figures and descriptions. After modification, for the data from 03:00 to 04:30, the red line indicates the lower boundary of the thin aerosol layer 3.2 km before cloud formation. For the data from 04:30 to 09:40, the black line indicates the cloud base height of the cloud layer. Due to the lack of clear meaning of the black line after 09:40, it is removed. The calculation methods for thin aerosol layer base and cloud base are both based on the differential zero crossing method, while for aerosol layer base, the lower boundary is identified by selecting an appropriate threshold for the differential signal of the echo signal.

The author has made modifications and added explanations for the boundary between the aerosol layer and the cloud layer in the original text's figures and descriptions, as following:

"According to Fig. 8(a), there are signals changing from weak to strong above the black curve near 3 km. After 5:00, the echo signal gradually increased and the laser could not penetrate, suggesting that this should be a process of cloud formation. The red and black lines in the figure correspond to the lower boundaries of the aerosol layer and cloud layer of interest, respectively, calculated by differential zero crossing method. According to the temperature and humidity profiles shown in Fig. 8(b), the temperature below 3.5 km is higher than 0°C, and the relative humidity reaches over 90% at 3 km-3.2 km. Therefore, it can be determined that the strong signal appearing near 3 km in the atmosphere is water cloud. "

Line 227: The backscattering coefficients at the different wavelengths are key for the retrieval scheme. So how do you exactly calculate the backscattering coefficient? In

cloudy situations, it is well known, that is quite difficult to retrieve the backscattering coefficient (either using Klett/Fernald or even the Raman method) because of the strong attenuation in the cloud layer, which does not allow the usage of a reference (aerosol-free) height to calibrate. The strong attenuation also makes the retrieval of the extinction a major issue. On the other hand, multiple scattering will take place in the clouds as soon as it gets densely enough. So how does your approach avoid the multiple-scattering effect?

Finally, how do these underlying uncertainties affect the retrieval of the microphysical parameters, such as the effective radius and cloud droplet number concentration?

**Answer:**

The reviewer is quite right that the accuracy of the backscatter coefficient is crucial for the retrieval of microphysical parameters, and its inversion error directly affects the final result.

In this manuscript, the backscattering coefficient at 355nm was retrieved using the Raman- Mie method. The inversion accuracy of this method is relatively high. The retrieval of 1064nm backscatter coefficient is performed using the Fernald method. We propose a multi-wavelength aerosol backscattering ratio parameterized-calibration method (MABP-CM) based on parameterized equations for reference condition calibration of Fernald inversion, which was described in detail in Ref. (Xinhong Wang, 2022).

The ground-based lidar is used for the observation of cloud in this manuscript. The laser divergence angle is less than 0.5mrad, the FOV of the telescope is 0.5mrad, and the detected cloud height is low. Therefore, the influence of multiple scattering on the results is weak and can be ignored.

More discussions and explanations about error analysis of the algorithm has be added to the revised draft.

Line 240: How is the maximum number concentration 130 cm-3? In the exemplary profile (Fig. 9f) there are no values larger than 100 cm-3. There was also no explanation of what was done below the cloud base. Was the retrieval version for aerosols applied here, or the cloud version was used for the whole profile?

**Answer:** The maximum number concentration is less than 100 $cm^{-3}$, and this will be revised in the revised manuscript. In the process of cloud shown in Figure 8, the aerosol hygroscopicity increase plays an important role. According to Figure 9b, the relative humidity reaches 100% near 3km, and below 3km, the relative humidity is less than 100%. Therefore, the aerosol lookup table is used below the cloud base for the retrieval of aerosol profiles, and the cloud droplet lookup table is used above the cloud base (gray shaded area).

Line 250 (Fig 10f): I do not see, how can it be possible, that the concentration of droplets in the cloud layer (~2000 cm-3), is two orders of magnitude than the aerosol concentration below the cloud (~10 cm-3). Where is all that CCN coming from?

**Answer:** We have carefully checked our results and ultimately determined that they are reasonable. But we need to pay attention to two points: 1) The number concentration (~2000 cm-3) pointed out by the reviewer is located near the altitude of 3.1km, where the inversion error of the data is the largest. The error range of the number concentration reaches (~200 cm-3 to~2000 cm-3), as indicated by the error bar in the manuscript; 2) There may be much CCN near this location. According to Figure 8, the cloud layer should be formed after the aerosol hygroscopicity increase. On that day, the vertical structure of the atmosphere had an uneven distribution of aerosols, and there should be more aerosols around 3km. And according to Figure 8 (b), it can also be observed that there is a significant inversion layer at 3.2km, so it is normal for there to be more aerosol accumulation below the inversion layer.

Line 258 (Fig. 12): Same issue as Fig 10. And why even bother defining regions 1 and 2, where there was no cloud at all yet?

**Answer:** The definitions of regions 1 and 2 in the figure are distinguished based on the echo signal in Figure 8. There are no clouds in regions 1 and 2, but based on the lidar echo signal, we can see a more obvious signal growth and change process. The echo signal in Figure 8 shows three distinct stages. The first stage (Region 1) is clouds have not yet formed, but an obvious layer of aerosol can be seen; In the second stage (region 2), there is a significant trend of enhancement in the echo signal; In the third stage (Region 3), cloud droplets are generated and clouds appear.

**Response to Anonymous Referee #3**

This study proposed an inversion method of atmosphere aerosol or cloud microphysical parameters based on dual wavelength lidar data. However, several comments below should be settled.

**Response:** Thank you very much for your nice comments. Your question and suggestion are very helpful for us to improve the quality of our paper. We appreciate the reviewer's thoughtful review and constructive comments. The following is our point-to-point replies.

1. Line 31: an5bd?

**Answer:** it is a clerical error. It should be "and". We will check and revise the manuscript sentence by sentence to avoid mistakes. Thanks.

2. Part 2.2: This section needs to be improved by expanding to include more supportive figures and detailed descriptions.

**Answer:** Thanks for your nice comments.

This section has been modified as

"In order to study the characteristics of APSDs and CDSDs in the vertical altitude, the APSDs and CDSDs obtained from aircraft observations by the Hebei Provincial Weather Modification Office were analyzed (from 2005 to 2006). APSDs were measured by the PCASP-100X probe, and CDSDs were obtained by the FSSP-100-ER probe [29]. The PCASP-100X is an optical particle counter for measuring aerosol size distribution from 0.10 μm to 3.00 μm in diameter in 15 different size bins with a frequency of 1 Hz. The sample flow volume in the PCASP-100X was set to 1 cm$^3$ s$^{-1}$. FSSP-100-ER is an instrument that measure cloud droplet size and concentration using light scattering, with the measurement range of 0.5-47 μm.

The obtained APSDs and CDSDs were fitted one by one using Gamma function. In order to minimize the error at all radius, the minimization problem is solved using the following equation

$$\int_0^{D_{max}} \left( \log\left( f_m(D) \right) - \log\left( f_{fitted}(D) \right) \right)^2 \mathrm{d}D \to \min \tag{11}$$

here, $f_m(D)$ is the actual particle size distribution measured by the PCASP-100X, $f_{fitted}(D)$ is the fitted distribution, $D$ is the aerosol particle diameter, $D_{max}$ is the measured maximum particle diameter. The goodness of fit $R^2$ is used to represent the difference between the fitting function and the measured data. The definition of goodness of fit is as follows:

$$R^2 = 1 - \frac{\sum_{i=1}^n \left( y_i - \hat{y}_l \right)^2}{\sum_{i=1}^n \left( y_i - \overline{y} \right)^2} \tag{12}$$

where $y_i$ is the measured value, $\hat{y}_l$ is the predictive value, $\overline{y}$ is the mean measured value. The numerator represents the sum of squared residuals, and the denominator represents the sum of squared total deviations.

~3500 sets of APSDs and 2221 sets of CDSDs were statistically analyzed. Over 95% of the data have a high goodness of fit in the Gamma distribution. The goodness of fit of

CDSDs is higher than that of APSDs, with CDSDs of 0.983 and APSDs of 0.856. The parameters of CDSDs are significantly larger than that of APSDs, and there are obvious differences of b and c for cloud and aerosol. The literature suggests that there is a certain functional relationship between the Gamma parameters b and c of CDSDs [30]. Statistical analysis was conducted on the b and c parameters of APSDs and CDSDs, as shown in Fig. 1.

[Figure]

Fig. 1 Statistical Results of parameter *b* and *c* in aerial survey data. (a)Aerosol particles, (b)cloud droplets.

According to Fig. 2, there are the remarkable linear relationships between parameter b and c. The fitting functions for CDSDs and APSDs are as follows:

$$\begin{cases} c_{\text{cloud}} = 0.33b_{\text{cloud}} + 0.60 \\ c_{\text{aerosol}} = 2.67b_{\text{aerosol}} + 7.43 \end{cases} \tag{13}$$

The linear relationship between the two parameters of CDSDs is better with a goodness of fit of 0.948, and a linear goodness of fit of 0.821 for APSDs. According to the statistical results, the parameter b of APSDs at vertical height is mainly distributed in the range of 2-7, and CDSDs is mainly distributed in the range of 2-8. "

3. Figure 2: Not clear, Add more details on the description of the algorithm. For example, the look-up-table, etc.

**Answer:** Thanks for your nice comments.

We have redrawed the flowchart of the algorithm, and shown as the following figure.

[Figure]

Figure I the algorithm flowchart for atmosphere particle microphysical parameters

And the descriptions of the algorithm are added.

"In this algorithm, the first step is to establish a lookup table between aerosol/cloud optical parameters and microphysical parameters. 1) Assuming that aerosol particles and cloud droplets follow the Gamma distributions, calculate the extinction coefficient and backscatter coefficient at different laser wavelengths (355nm and 1064nm in this paper) based on the Mie scattering theory; 2) Calculate the ratio of backscatter coefficients for two wavelengths, which is the backscatter color ratio, or calculate the ratio of extinction coefficient to backscatter coefficient, which is the radar ratio; 3) Change the parameters of the aerosol to obtain the gamma distributions with effective radius from 0.2 μm to 3 μm, calculate the optical parameters and corresponding optical parameter ratios (radar ratio or backscatter color ratio) for each Gamma distribution, and establish the lookup table for aerosol effective radius; 4) Similar to the step 3, establish the lookup table for cloud drops (effective radius are from 0.5 μm to 5 μm). After the lookup table is completed, the microphysical parameters of aerosols or clouds are calculated based on the lookup tables and LiDAR detection data. The specific steps are as follows: 1) the dual-wavelength (355 nm and 1064 nm) Raman LiDAR need be selected for the detection of atmosphere; 2) Raman and Fernald methods are used for the retrieval of optical parameters at multi-wavelengths, and the backscatter color ratio or lidar ratio can be obtained ; 3) aerosol and cloud layers are identified based on lidar echo signals; 4) Retrieve the effective radius of aerosols or cloud droplets at different heights based on optical parameters ratios and lookup tables; 5) Calculate the parameters b and c in the Gamma distribution according to formulas (13) and (16); 6) Calculate the value of a in the Gamma distribution according to formula 17; 7) Calculate the number concentration according to formula (3). "

4. line 136: How to determine the boundary of the blue box?

**Answer:** The boundary of the blue box is determined by the common monotonic variation interval of multiple curves. Thanks.

5. line 143: you claimed that the larger the value of b, the more pronounced the Gamma function describes the characteristics of large particles. Why did you choose b=6 for cloud droplets, and b=3 for aerosols?

**Answer:** Because the particle size of cloud droplets is larger than that of aerosols, it is preferred to choose the b-value of cloud droplet as the larger value and the aerosol as the smaller value. On the other hand, according to Figures 4 and 5, we can also observe that the influence of b value on the results is not significant. In addition, only the initial value of b was selected here, and b will be recalculated when using the Look up Table for subsequent calculations.

6. Figure 5: these figures do not match the description. The 3.2.2 part should be modified.

**Answer:** "when the complex refractive index of particles changes, the color ratio curves will fluctuate, but they always monotonically decreases at 0.3 μm to 1.7 μm." has been modified to "when the complex refractive index of particles changes, the color ratio curves will fluctuate, but they always monotonically decreases from 0.3 μm to 1 μm." Thanks.

7. Line 159-160: It is claimed that "According to Fig. 5, when the complex refractive index of particles changes, the color ratio curves will fluctuate, but they always monotonically decrease at 0.3 μm to 1.7 μm.". However, the content displayed in some figures does not align with the aforementioned description. Furthermore, why is there a need to emphasize "0.3 μm to 1.7 μm"? How do the authors determine these two boundary values?

**Answer:** This question is the same as the previous one. There is an error in the manuscript,and it should be "they always monotonically decrease at 0.3 μm to 1 μm. " this error will be modified in the revised manuscript. Thanks.

8. Figure 6: The text of the legend of Figure6(a) and 6(b) such as Inversion value and True value should be modified.

**Answer:** Thanks

   "True value" has been modified to "the initial values" , and "Inversion value" has been modified to "the retrieved values". And the Figure 6 are shown as:

[Figure]

**Figure 6**

9.  Part 3.3, line 182-183: It is claimed that the first three factors have been discussed earlier and this section focuses on the inversion error introduced by optical parameters. As the error analysis of the algorithm, the all factors which affected the inversion algorithm should be discussed.

**Answer:** Yes, the opinion of reviewer is right that "all factors which affected the inversion algorithm should be discussed". More discussions and explanations about error analysis of the algorithm will be added to the revised draft.

For urban aerosols and water clouds, their particles are spherical, so the error caused by non-spherical particles can be ignored.

The error introduced by the assumption of Gamm distribution is relatively complex and difficult to accurately calculate. This study evaluates this error by numerical simulation based on APSDs and CDSDs data by aircraft observations. Actually, the error presented in Figure 7 is mainly caused by the assumption of Gamm distribution. Calculate optical parameters of over 5000 sets of APSDs and CDSDs data, and retrieve the microphysical parameters using our algorithm. The calculated standard deviations between the inversion results and the actual data are: for aerosols, the standard deviation of the effective radius is ~10%, and the standard deviation of numerical concentration is 20%; For clouds, the standard deviation of the effective radius is 15%, and the standard deviation of numerical concentration is 20%.

The deviation introduced by improper assumption of complex refractive index may be the largest term in this technique. For water clouds, the complex refractive index is stable and the deviation caused by it can be ignored. It is difficult to accurately obtain the complex refractive index of aerosols, and the deviation caused by the complex refractive index may reach over 100%. Figure 6 shows the effect of complex refractive index

variation on the optical parameter ratio. From Figure 6, it can be seen that when the real part of the complex refractive index changes within the range of 0.03 and the imaginary part changes within 0.01, the effective radius deviation caused by the complex refractive index is within a controllable range. After calculation, the deviation does not exceed 40%. And it can be seen that although complex refractive index can lead to the significant change of the effective radius value, when the aerosol is constant, its monotonic characteristics remain unchanged, which means that the evaluation of particle size changes is reliable.

The above three errors are independent of each other. Considering the actual inversion ability of LiDAR, the deviation of color ratio will reach 10%. The final evaluation shows that the mean square deviation of the inversion error of aerosol effective radius is less than 45%, and the standard deviation of the inversion error of cloud droplet effective radius is 25%.

10. Part 4.2: the experimental observation of a cloud generation process was provided in this part. How can the experiment be confirmed as a test of the cloud generation process, rather than clouds drifting in from other locations?

**Answer:**

When using LiDAR for vertical detection at fixed points, it often detects clouds floating over from other places. We checked the wind speed data of the sounding balloon that day, and found that the wind speed was 9m/s at 3 km. Therefore, the cloud layer we detected should drift in from other locations. We have observed the microphysical changes of the cloud lateral boundary.

[Figure]

11. Figure 8: The description of the Figure 8 should be modified. Figure 8(b) is not the lidar observations.

**Answer:** Thanks,Figure 8(b) is not the lidar observations, and it is the temperature and relative humidity profiles obtained from the sounding balloon at 7:15 am (BJT). The description of the Figure 8 has been modified in the revised draft. And they are "Lidar and sounding balloon observations. (a) THI diagram of RSCS at 1064 nm observed by

lidar at 03:00-10:00 September 16, 2022(CST), (b) temperature and relative humidity observed by sounding balloon at 07:15 September 16, 2022."

12. Figure 9 and Figure 10: What the error bars stand for?

**Answer:** The error bar represents the uncertainty of the inversion result. The error of backscattering coefficient and backscatter color ratio is determined by the signal-to-noise ratio of the LiDAR system and the error of the optical parameter inversion algorithm. The error bar of the effective radius represents the uncertainty of the results caused by optical parameter errors and gamma distribution assumption errors.

13. Only one experimental observation was provided. Could you provide more experiments?

**Answer:** Our experimental observation data has other cases, as shown in the following figures, which is another case of observed cloud formation process. The variation process of effective radius and numerical concentration can be clearly displayed. However, this article focuses on the research of data inversion methods, so one case in the article is sufficient. Thanks!

[Figure]

Fig. I Lidar observations at 20:00 June 7 to 4:00 June 8, 2022(CST) (THI diagram of RSCS at 1064 nm)

[Figure]

[Figure]

Fig. II Microphysical parameters inversion results of atmospheric particulate matters at 20:00 June 07-04:00 June 08, 2021(CST). (a)Effective radius, (b)number concentration